# DOUBT: Decoupled Object-level Understanding and Bridging via vMF-based Trustworthiness for Hallucination Detection in MLLMs

Kaiqi Chen [1]  Yang Qin [1]  Changhao He [1]  Xi Peng [2 3]  Peng Hu [* 1]

## Abstract

Multimodal Large Language Models (MLLMs) frequently produce hallucinations (i.e., assertions that contradict the image or facts), undermining reliability in high-risk applications. Existing detection approaches typically feed images and texts jointly and estimate hallucination scores by measuring the consistency of model outputs. However, because the visual module often lags behind the language module in understanding and reasoning, MLLMs can repeatedly produce similar yet incorrect answers, yielding overestimated trustworthiness and missed detections. To address this, we propose a simple yet effective model-agnostic method, dubbed **D**ecoupled **O**bject-level **U**nderstanding and **B**ridging via vMF-based **T**rustworthiness (DOUBT). DOUBT first employs Object-level Understanding and Bridging (OUB), a two-step prompting scheme that decouples object recognition from relational reasoning by prompting the model to identify objects and then reason based on them. It further introduces a von Mises-Fisher (vMF)-based trustworthiness metric, which is more stable than semantic entropy metrics in small-sample settings. Extensive experiments and ablation studies on multiple benchmarks show that DOUBT consistently outperforms state-of-the-art baselines, demonstrating its robustness and generalizability for hallucination detection in MLLMs. The code is available at https://github.com/XLearning-SCU/2026-ICML-DOUBT.

---

[1]College of Computer Science, Sichuan University, Chengdu, China [2]School of Artificial Intelligence, Sichuan University, Chengdu, China [3]National Key Laboratory of Fundamental Algorithms and Models for Engineering Numerical Simulation, Sichuan University, Chengdu, China. Correspondence to: Peng Hu <penghu.ml@gmail.com>.

*Proceedings of the $43^{rd}$ International Conference on Machine Learning*, Seoul, South Korea. PMLR 306, 2026. Copyright 2026 by the author(s).

## 1. Introduction

In recent years, Multimodal Large Language Models (MLLMs) have achieved remarkable progress and shown strong performance on various tasks such as visual question answering and multimodal object reasoning (Li et al., 2023a; Wang et al., 2024b; Qin et al., 2026; Feng et al., 2023b). Despite these advances, MLLMs still frequently generate assertions that contradict input images or external facts, commonly referred to as **hallucination** (Bai et al., 2024; Liu et al., 2024c; Huang et al., 2024). Such hallucinations undermine the trustworthiness of MLLM outputs and can pose serious safety risks in high-stakes domains (Feng et al., 2023a), including medical diagnosis, embodied intelligence, legal analysis, scientific discovery, and autonomous driving (Ji et al., 2023; Bubeck et al., 2023; Feng et al., 2026a;b; 2025; Jiang et al., 2025; Li et al., 2025b). As MLLMs are increasingly deployed in real-world applications, hallucination detection has become essential for safe and trustworthy AI (OpenAI, 2023).

Existing approaches for hallucination detection can be broadly categorized into **white-box** and **black-box** approaches (Huang et al., 2025; Ahadian & Guan, 2025). White-box methods inspect model-internal signals (e.g., attention, activations, gradients) to infer hallucination scores, but require access to the model architecture, which is often infeasible for closed-source models (Dasgupta et al., 2025; Li et al., 2025a). Black-box methods, in contrast, operate solely on observable outputs instead of model-internal signals, such as by verifying model outputs against external knowledge bases or by sampling multiple responses to estimate output diversity (Manakul et al., 2023; Li et al., 2024). Although black-box methods are flexible and broadly deployable, most of them often rely on complete external knowledge bases and stable metrics, limiting their robustness in practice. To get rid of such dependence, recent approaches employ MLLMs to generate sufficiently diverse answers for uncertainty estimation in hallucination detection (Chen et al., 2024a), thereby providing a generalizable and easily deployable solution.

Although the uncertainty-based methods (Farquhar et al., 2024) are effective at detecting hallucinations, they face two major challenges in practice. First, MLLMs tend to

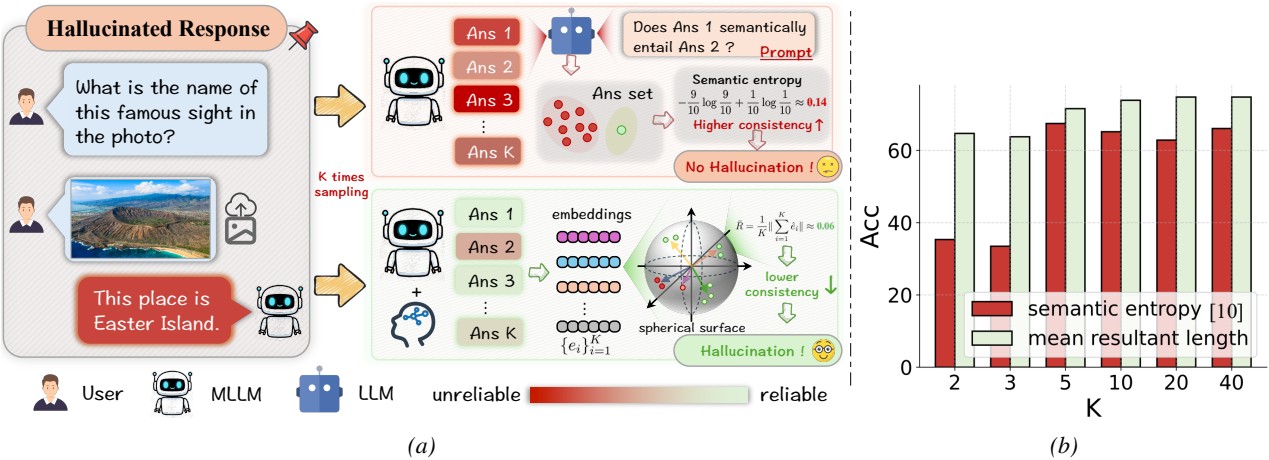

*Figure 1.* (a) Comparison of two ways to elicit model responses. If we directly query the model, it may generate many hallucinated answers with low uncertainty, leading to detection failure. In contrast, if we guide the model's responses, it is more likely to produce correct answers, making the uncertainty estimation more accurate. (b) Accuracy comparison of the two metrics under the same DOUBT framework. When the sample size $K$ is small, the performance of semantic entropy (Farquhar et al., 2024) drops significantly and becomes highly unstable.

repeatedly generate semantically similar but incorrect answers when faced with object hallucinations, producing low measured uncertainty and missed detections as shown in Figure 4. This stems from the modality gap where the visual module often lags behind the LLM's reasoning capabilities, leading to object-logic mismatches (Zhai et al., 2023). Second, existing methods commonly compute cluster-level semantic entropy to quantify the uncertainty scores by clustering multiple responses based on semantics and measuring cluster probabilities (Zhang et al., 2024; Kossen et al., 2024). However, semantic entropy is sensitive to clustering and unstable when the sample size $K$ is small, as shown in Figure 1b.

To address the above limitations, we propose a model-agnostic approach that focuses on object-level hallucination detection, termed **D**ecoupled **O**bject-level **U**nderstanding and **B**ridging via vMF-based **T**rustworthiness (DOUBT). Our DOUBT proceeds in two stages. In the first stage, we address the limitations of the visual module by decoupling object recognition and relational understanding through a step-by-step reasoning paradigm, encouraging models to produce more diverse and accurate responses. More specifically, an MLLM is first prompted to i) list recognized objects in the input image, and ii) reason about the image conditioned on these objects using a Chain-of-Thought (CoT) prompt (Wei et al., 2022) to obtain new responses. In the end, we can sample the responses through both direct sampling and object-guided reasoning using the same input image to enhance the diversity of the response space, facilitating hallucination detection with only prompting strategies rather than modifying model architectures. In the second stage, to overcome the limitations of traditional semantic entropy in better assessing the uncertainty in the response set, we draw inspiration from the von Mises–Fisher (vMF)

distribution (Fisher, 1953) and measure it using the mean resultant length, which is simple yet efficient to compute and can directly measure the similarity of input samples based on the geometric properties of the feature space. It proves to be more robust under the same conditions in Figure 1b. Once computed, we take the average uncertainty of the two types of responses and compare it with a predefined threshold to detect hallucinations.

Our main contributions are summarized as follows:

- This work reveals that the reasoning limitations of the visual module could cause the output results to remain incorrectly consistent, thus degrading the performance of hallucination detection. To address this, we decouple simple recognition from hard reasoning, and then guide the model to elicit richer and object-aware responses, thus improving hallucination detection.

- We propose a model-agnostic uncertainty metric based on the vMF distribution to alleviate the instability of traditional entropy measurement, which can be seamlessly integrated as a plug-and-play module into existing hallucination detection pipelines.

- We present extensive experiments across multiple MLLMs, scales, and four widely-used benchmarks (i.e., LLaVABench, MMVet, MMMU, and ScienceQA), along with ablation studies demonstrating the superior accuracy and stability of our DOUBT.

## 2. Related Work

### 2.1. Hallucination Detection

Hallucination detection has attracted increasing attention from academia and industry as MLLMs are deployed in safety-critical scenarios (Sahoo et al., 2024; Moor et al., 2023). Existing methods can be roughly grouped into three categories. i) Internal-signal methods analyze model-internal signals, such as activations, attentions, logits, or gradient-based uncertainty, to infer whether a response is hallucinated (Kadavath et al., 2022; Suresh et al., 2025). Although these methods can be effective, they require access to model internals, which is often inaccessible for proprietary or closed-source MLLMs (Dasgupta et al., 2025). ii) External-knowledge methods verify responses using external knowledge sources or fact-checking modules, ensuring consistency with objective information (Choi et al., 2023; Zhang et al., 2025a). These methods are often reliable when relevant knowledge exists, but they incur additional computational overhead and depend on the completeness and quality of external resources (Sok et al., 2025). iii) Consistency-based methods detect hallucinations by comparing multiple model outputs, based on the intuition that correct answers tend to be self-consistent while hallucinations vary (Zhang et al., 2023; Kossen et al., 2024). Although broadly applicable and model-agnostic, these approaches can still fail when an MLLM repeatedly generates consistent but incorrect responses, particularly when the visual understanding module is weak (Chen et al., 2024a; Srey et al., 2025). In such cases, uncertainty appears seemingly low, and hallucinations remain undetected. To address this issue, this work proposes an object-level understanding and bridging approach to diversify and improve sampled responses, thereby ensuring that uncertainty estimates more accurately reflect true reliability.

### 2.2. Uncertainty Estimation

As a key tool for assessing model confidence and reliability, uncertainty estimation has emerged as an important topic within MLLM research (Kendall & Gal, 2017; He et al., 2023). It plays an important role in applications where robustness and trustworthiness are essential (He et al., 2026; 2024; Su et al., 2025; 2026). In the context of hallucination detection for MLLMs, uncertainty provides a practical proxy for whether model outputs should be trusted (Ovadia et al., 2019; Nguyen et al., 2025). Existing approaches for measuring uncertainty can be broadly divided into two main categories. Distribution-based methods rely on the predictive probability distribution of the model, with entropy and mutual information being the most common measures (Gal & Ghahramani, 2016; Lakshminarayanan et al., 2017). These methods capture how concentrated or dispersed the output distribution is, but they often fail when models are overcon-

fident in wrong predictions. Sampling-based methods, on the other hand, estimate uncertainty by generating multiple outputs and assessing their diversity (Kuleshov et al., 2018; Tian et al., 2025). Among these, semantic entropy (Kossen et al., 2024; Nikitin et al., 2024) has been widely used in this category, where the variance across responses is taken as a signal of uncertainty. Despite their effectiveness, entropy-based methods are highly sensitive to the number of samples and can be unstable in low-sample regimes. To address this, we propose a vMF-inspired trustworthiness metric that provides a smoother and more sample-efficient approximation of uncertainty, especially in low-sample regimes.

## 3. Method

### 3.1. Problem Statement

This work studies hallucination detection in a black-box setting, where no model internals or external knowledge bases are accessible. For the $i$-th multimodal input $p_i = \{I_i, t_i\}$, where $I_i$ is the image and $t_i$ is the task instruction (e.g., question or caption), an MLLM produces a response $A = \text{MLLM}(p)$. Following prior work (Farquhar et al., 2024), we first obtain a deterministic reference answer $A_i = \text{MLLM}(p_i)$ using a low temperature. To assess its reliability, we then query the same input $K$ times: $s_{\text{ori}} = \{a_1, a_2, \ldots, a_K\}$, which forms a response set used to estimate uncertainty for hallucination detection. However, if the model exhibits limited visual understanding, it may repeatedly generate consistent but incorrect answers, leading to falsely low estimated uncertainty.

To mitigate this, we introduce an Object-level Understanding and Bridging (OUB) mechanism that prompts the model to explicitly decouple object recognition (i.e., understanding) from relational reasoning (i.e., bridging) during inference. This process yields an additional answer set $s_{\text{bri}}$, which not only enhances the model's understanding capability but also increases response diversity, providing a contrastive reference for evaluation (Section 3.2). After obtaining $s_{\text{ori}}$ and $s_{\text{bri}}$, we estimate the trustworthiness score of each response set using a vMF-based trustworthiness metric and compute their average (Section 3.3). The average score is then compared with a predefined threshold $\theta$ to determine whether the model's response is reliable, thereby enabling effective hallucination detection. To make the overall procedure clearer, we summarize the complete algorithmic pipeline of DOUBT in Algorithm 1.

### 3.2. Object-level Understanding and Bridging

To improve the reasoning ability of the model, we design a two-stage prompting scheme that produces a set of bridging responses $s_{\text{bri}}$.

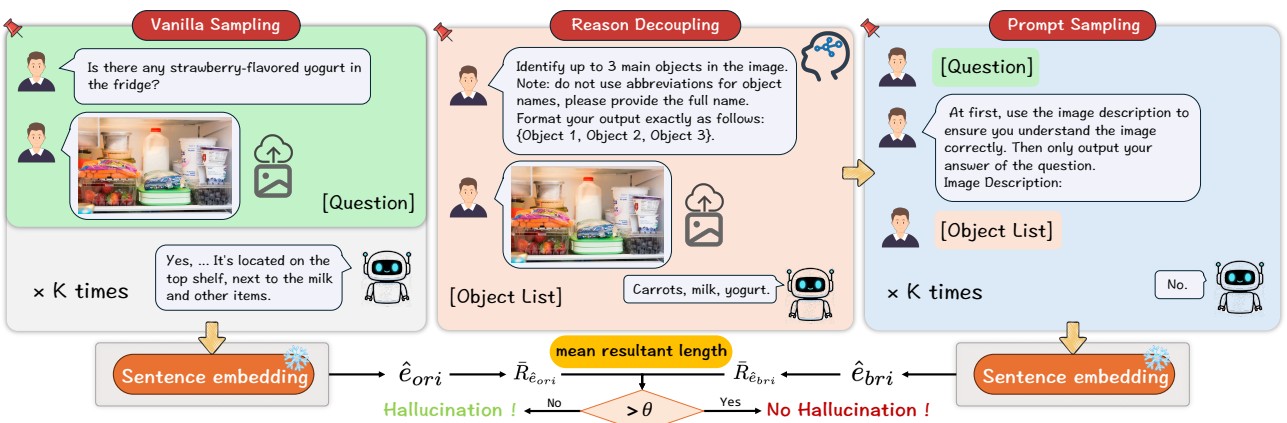

*Figure 2.* Overview of our DOUBT. Given an input, we obtain two response sets: direct responses and object-guided responses via decoupled prompting. Each set is embedded and scored using our vMF-based trustworthiness metric. The average score is then compared to a threshold to determine hallucination.

---

**Algorithm 1** Algorithmic Pipeline of DOUBT for Hallucination Detection

---

**Require:** Image $I_i$, task instruction $t_i$, MLLM $\mathcal{M}$, object recognition prompt $p_{\text{obj}}$, bridging prompt $p_{\text{bri}}$, sampling number $K$, threshold $\theta$

**Ensure:** Detection flag $flag$

1: Obtain the reference answer $A_i \leftarrow \mathcal{M}(I_i, t_i)$.
2: Sample direct responses $s_{\text{ori}} = \{a_k\}_{k=1}^{K}$ from $\mathcal{M}(I_i, t_i)$.
3: Obtain object list $r_i \leftarrow \mathcal{M}(I_i, p_{\text{obj}})$.
4: Sample object-bridged responses $s_{\text{bri}} = \{b_k\}_{k=1}^{K}$ from $\mathcal{M}(I_i, t_i \| p_{\text{bri}} \| r_i)$.
5: Encode and normalize $s_{\text{ori}}$ and $s_{\text{bri}}$ into $\hat{E}_{\text{ori}}$ and $\hat{E}_{\text{bri}}$.
6: Compute $\bar{R}_{\hat{E}_{\text{ori}}}$ and $\bar{R}_{\hat{E}_{\text{bri}}}$ by the mean resultant length.
7: Compute $\bar{R}_{\text{avg}} = \frac{1}{2}(\bar{R}_{\hat{E}_{\text{ori}}} + \bar{R}_{\hat{E}_{\text{bri}}})$.
8: **if** $\bar{R}_{\text{avg}} > \theta$ **then**
9:    $flag \leftarrow 1$ {Non-hallucinatory}
10: **else**
11:    $flag \leftarrow 0$ {Hallucination}
12: **end if**
13: **return** $flag$

---

**Stage 1: Object-level Understanding** We begin with an object recognition prompt $p_{\text{obj}}$ that instructs the model to identify the salient visual entities in the image. Objects serve as the fundamental building blocks of visual semantics, which provide the primary cues for understanding what the scene contains and guiding reasoning about the image. In other words, the correlations among these prominent objects generally determine the overall meaning of the image, guiding subsequent reasoning about actions, attributes, and higher-level contextual relationships. Establishing an accurate inventory of objects through $p_{\text{obj}}$ is therefore critical for grounded reasoning. We empirically limit $p_{\text{obj}}$ to 'up to 3 main objects' to balance attention focus with coverage. An ablation study on the maximum number of recognized objects is provided in Appendix E. The prompt $p_{\text{obj}}$ is shown as follows:

*"Identify up to 3 main objects in the image. Note: do not use abbreviations for object names, please provide the full name. Format your output exactly as follows: {Object 1, Object 2, Object 3}."*

Mathematically, the object recognition result $r_i$ is then obtained as:

$$r_i = \text{MLLM}(\{I_i, p_{\text{obj}}\}). \qquad (1)$$

**Stage 2: Object-level Bridging** Next, we design a bridging prompt $p_{\text{bri}}$ that guides the model to reason about the image conditioned on the recognized objects, and then generate an answer. This step encourages the model to verify and refine its understanding based on explicit object information, thereby reducing the likelihood of generating spurious or hallucinated responses. The full bridging prompt $p_{\text{bri}}$ is given as:

*"At first, use the image description to ensure you understand the image correctly. Then only output your answer of the question. \nImage Description: {Object 1, Object 2, Object 3}."*

With $p_{\text{bri}}$, we prompt the model to generate $K$ object-bridged responses to form the bridging answer set $s_{\text{bri}}$ as below: $s_{\text{bri}} = \{b_1, b_2, \dots, b_K\}$, where $b_i = \text{MLLM}(\{I_i, t_i \| p_{\text{bri}} \| r_i\})$.

Through this two-step prompting, the model not only follows the user's instruction but also anchors reasoning on explicit object-level semantics, thus expanding the diversity and reliability of the answer space, facilitating trustworthy hallucination detection. We note that DOUBT relies on the model's object recognition capability, and severe object misidentification may propagate to later stages. However, since both stages are driven by the same MLLM, such failures typically manifest as increased response inconsistency,

which our trustworthiness metric is designed to capture. Regarding complexity, while the two-step prompting linearly increases inference cost, it eliminates the substantial training overhead of white-box methods, making it a cost-effective trade-off for high-stakes black-box deployment.

### 3.3. vMF-Based Trustworthiness

A key indicator of hallucinations is the semantic divergence among multiple responses: dispersed answers suggest ambiguity in the model's understanding and thus a higher risk of hallucination. However, traditional entropy-based measures heavily rely on the accuracy of semantic clustering, which is unstable under small sample sizes. To overcome this, we present a geometric trustworthiness metric inspired by the vMF distribution.

The probability density function of the vMF distribution is:

$$p(x|\mu,\kappa) = C_d(\kappa)\exp(\kappa\mu^T x), \tag{2}$$

where $x \in \mathbb{R}^d$ is a unit vector, $\mu \in \mathbb{R}^d$ is the mean direction with $\|\mu\| = 1$, $\kappa \geq 0$ is the concentration parameter, and $C_d(\kappa)$ is a normalization constant in $d$ dimensions.

On the unit spherical surface, response embeddings following a vMF distribution exhibit concentration governed by $\kappa$. Since $\kappa$ is unbounded and difficult to threshold directly, we instead use the sample mean resultant length $\bar{R} = \frac{1}{K}\|\sum_{i=1}^{K}\hat{x}_i\|$, which is an unbiased estimate related to $\kappa$ according to the directional statistics theory (Mardia & Jupp, 2009):

$$\bar{R} = A_d(\kappa) = \frac{I_{d/2}(\kappa)}{I_{d/2-1}(\kappa)}, \tag{3}$$

where $A_d(\kappa)$ denotes the ratio of modified Bessel functions that characterizes the mean resultant length in $d$-dimensional space; $I_\nu(\cdot)$ is the modified Bessel function of the first kind. Like $\kappa$, $\bar{R}$ can be directly used to measure the concentration of the response distribution. Specifically, a larger $\bar{R}$ is equivalent to a larger $\kappa$ in the vMF distribution, indicating higher response consistency and lower hallucination risk.

To compute $\bar{R}$, we encode each textual response using the `nli-roberta-large` model (Reimers & Gurevych, 2019), yielding embedding sets: $E_{\text{ori}} = \{e_{a_1}, e_{a_2}, \ldots, e_{a_K}\}$ and $E_{\text{bri}} = \{e_{b_1}, e_{b_2}, \ldots, e_{b_K}\}$, where $e_{a_i} = \text{nrl}(a_i)$ and $e_{b_i} = \text{nrl}(b_i)$. Since the vMF distribution is defined over directional data, i.e., points lying on the surface of a unit hypersphere, we normalize $E_{\text{ori}}$ and $E_{\text{bri}}$ onto a unit hypersphere as follows: $\hat{E}_{\text{ori}} = \{\hat{e}_{a_i}\}_{i=1}^{K} = \{\frac{e_{a_i}}{\|e_{a_i}\|}\}_{i=1}^{K}$ and $\hat{E}_{\text{bri}} = \{\hat{e}_{b_i}\}_{i=1}^{K} = \{\frac{e_{b_i}}{\|e_{b_i}\|}\}_{i=1}^{K}$. We normalize embeddings onto a unit hypersphere because, in high-dimensional semantic spaces (e.g., RoBERTa (Liu et al., 2019)), semantic similarity is primarily encoded in direction (angular distance) rather than magnitude. Thus, vMF

offers a more theoretically grounded measure for semantic consistency than Euclidean-based metrics.

After normalization, the mean resultant lengths for the two response sets (i.e., $\hat{E}_{\text{ori}}$ and $\hat{E}_{\text{bri}}$) are respectively computed as: $\bar{R}_{\hat{E}_{\text{ori}}} = \frac{1}{K}\|\sum_{i=1}^{K}\hat{e}_{a_i}\|$ and $\bar{R}_{\hat{E}_{\text{bri}}} = \frac{1}{K}\|\sum_{i=1}^{K}\hat{e}_{b_i}\|$. Finally, we average them to obtain the overall trustworthiness score:

$$\bar{R}_{\text{avg}} = \frac{1}{2}(\bar{R}_{\hat{E}_{\text{ori}}} + \bar{R}_{\hat{E}_{\text{bri}}}). \tag{4}$$

A higher $\bar{R}_{\text{avg}}$ indicates greater internal consistency among responses and therefore higher confidence in the model's reliability.

### 3.4. Hallucination Detection Criterion

We determine the presence of hallucination by comparing the average trustworthiness score $\bar{R}_{\text{avg}}$ against a given threshold $\theta$. Formally, the hallucination flag is given by:

$$flag = \begin{cases} 1, & \text{if } \bar{R}_{\text{avg}} > \theta, \\ 0, & \text{if } \bar{R}_{\text{avg}} \leq \theta, \end{cases} \tag{5}$$

where $flag = 1$ denotes that the model's responses exhibit sufficiently high internal trustworthiness and are therefore judged as non-hallucinatory, while $flag = 0$ denotes low trustworthiness and is treated as a hallucination case. This decision rule aligns with the intuition that reliable answers should be directionally coherent across multiple reasoning paths, whereas hallucinations typically result in semantic divergence.

Finally, we evaluate detection accuracy by comparing the reference answer $A_i$ against its ground-truth answer $gt_i$. More specifically, we compare the predicted hallucination flag with the ground-truth correctness of the reference answer $A_i$:

$$Accuracy = \frac{1}{N}\sum_{i=1}^{N}\mathbb{I}\left[\mathbb{I}[A_i = gt_i] = flag\right], \tag{6}$$

where $\mathbb{I}[\cdot]$ denotes the indicator function and $N$ is the total number of evaluation samples. Generally, a detection is considered correct if the predicted hallucination flag aligns with the actual correctness of the answer, and the overall accuracy measures the proportion of correctly detected samples.

## 4. Experiment

### 4.1. Experimental Setup

**Datasets.** We evaluate our method on widely used benchmarks covering both free-form and multiple-choice settings. For free-form evaluation, we use LLaVABench (Liu et al., 2023) and MMVet (Yu et al., 2024), which assess high-level visual reasoning and integrated vision-language understanding. For multiple-choice evaluation, we further

| Dataset | Method | Q2B | Q7B | Q72B | I1B | I8B | I26B | L7B | L13B | LN7B | LN13B | Avg |
|---|---|---|---|---|---|---|---|---|---|---|---|---|
| LLaVABench | GAVIE (2024) | 25.00 | 26.67 | 40.00 | 30.00 | 31.67 | 31.67 | 15.00 | 20.00 | 45.00 | 35.00 | 30.00 |
| | Semantic Entropy (2024) | 61.67 | 55.00 | _61.67_ | _65.00_ | _60.00_ | 53.33 | 70.00 | **70.00** | 61.67 | **65.00** | 62.33 |
| | KLE (2024) | 28.33 | 48.33 | 58.33 | 40.00 | 46.67 | 50.00 | 23.33 | 45.00 | 43.33 | 33.33 | 41.67 |
| | EigenScore (2024) | _63.33_ | **58.33** | **63.33** | 63.33 | 55.00 | **61.67** | 68.33 | **70.00** | **70.00** | **65.00** | _63.83_ |
| | VL-Uncertainty (2025) | 56.67 | 53.33 | 53.33 | 60.00 | _60.00_ | 51.67 | _73.33_ | 63.33 | 61.67 | _61.67_ | 59.50 |
| | SNNE (2025) | 45.00 | _56.67_ | 55.00 | 38.33 | 53.33 | _56.67_ | 31.67 | 41.67 | 41.67 | 46.67 | 46.67 |
| | **Ours** | **68.33** | **58.33** | 53.33 | **73.33** | **61.67** | 55.00 | **80.00** | _66.67_ | _63.33_ | **65.00** | **64.50** |
| MMVet | GAVIE (2024) | 29.36 | 43.58 | 51.38 | 30.73 | 30.73 | 22.48 | 23.39 | 24.77 | 37.61 | 43.58 | 33.76 |
| | Semantic Entropy (2024) | 60.55 | 57.80 | 62.84 | 72.94 | 55.05 | 58.72 | 72.48 | _79.36_ | 61.01 | 72.48 | 65.32 |
| | KLE (2024) | 45.41 | 51.83 | 56.88 | 42.20 | 46.79 | 51.38 | 41.74 | 41.28 | 41.74 | 42.66 | 46.19 |
| | EigenScore (2024) | **73.85** | _70.64_ | **72.48** | _77.98_ | _67.43_ | **76.61** | 73.85 | 78.44 | _73.85_ | _77.52_ | _74.27_ |
| | VL-Uncertainty (2025) | _64.22_ | 67.43 | 66.97 | 65.60 | 62.39 | 64.67 | **79.35** | **80.28** | 66.06 | 69.72 | 68.67 |
| | SNNE (2025) | 48.62 | 57.80 | 63.30 | 40.37 | 49.54 | 54.13 | 39.45 | 42.20 | 43.12 | 48.62 | 48.72 |
| | **Ours** | **73.85** | **72.94** | _71.10_ | **78.44** | **72.02** | _75.23_ | _76.61_ | 77.98 | **76.15** | **77.98** | **75.23** |
| MMMU | GAVIE (2024) | 37.82 | 48.36 | 57.09 | 40.61 | 48.12 | 33.21 | 37.58 | 44.61 | 43.64 | 45.82 | 43.69 |
| | Semantic Entropy (2024) | 53.82 | 54.91 | 60.36 | 53.82 | 54.91 | 52.48 | 52.61 | 50.18 | 52.61 | 50.18 | 53.59 |
| | KLE (2024) | 43.88 | 53.33 | 62.91 | 46.42 | 49.58 | _59.52_ | 51.03 | 45.33 | 47.39 | 51.52 | 51.09 |
| | EigenScore (2024) | 52.85 | _64.48_ | _67.64_ | 51.15 | **59.39** | 56.24 | **59.03** | 54.42 | 55.39 | 49.21 | 56.84 |
| | VL-Uncertainty (2025) | _57.33_ | 58.55 | 65.94 | _55.15_ | _57.33_ | 57.21 | 56.36 | _55.15_ | _57.58_ | **56.24** | _57.68_ |
| | SNNE (2025) | 43.52 | 53.33 | 62.79 | 48.24 | 47.27 | 59.27 | 40.73 | 49.58 | 45.94 | _55.52_ | 50.62 |
| | **Ours** | **60.85** | **64.61** | **68.72** | **57.58** | **59.39** | **59.64** | _58.55_ | **56.48** | **60.85** | _55.52_ | **60.22** |
| ScienceQA | GAVIE (2024) | 61.82 | 77.09 | 85.23 | 53.94 | 86.71 | 89.19 | 58.50 | 66.39 | 62.27 | 65.20 | 70.63 |
| | Semantic Entropy (2024) | 54.04 | 77.94 | 87.06 | 64.45 | **90.08** | 91.32 | 61.77 | 68.02 | 67.67 | 65.34 | 72.77 |
| | KLE (2024) | 62.22 | 76.45 | 86.91 | _67.63_ | 89.64 | 90.43 | 60.73 | 67.97 | 65.94 | 66.23 | 73.42 |
| | EigenScore (2024) | 62.57 | 72.73 | 84.63 | 62.22 | 57.51 | 87.31 | 64.15 | 65.64 | _70.80_ | 64.95 | 69.25 |
| | VL-Uncertainty (2025) | **66.83** | **80.71** | _88.60_ | 64.50 | _89.54_ | **91.57** | _65.79_ | _68.57_ | 68.67 | **67.67** | _75.25_ |
| | SNNE (2025) | 64.35 | 78.63 | 84.68 | 66.04 | 79.18 | 83.14 | 65.54 | 68.02 | 70.70 | _67.50_ | 72.78 |
| | **Ours** | _65.00_ | _79.87_ | **88.80** | **68.32** | **90.08** | _91.37_ | **66.48** | **72.43** | **70.95** | 67.43 | **76.07** |

*Table 1.* Comparison between our method and other state-of-the-art baselines. Q, I, L, and LN denote Qwen2VL, InternVL2, LLaVA-1.5, and LLaVA-NeXT, respectively. The reported results are detection accuracies in percentage. **Bold** numbers indicate the best performance, while underlined numbers represent the second best.

adopt MMMU (Yue et al., 2024), a challenging college-level benchmark, and ScienceQA (Lu et al., 2022), which evaluates factual knowledge and systematic reasoning.

**Models.** We evaluate ten representative MLLMs from four architectural families: Qwen2VL, InternVL2, LLaVA-1.5, and LLaVA-NeXT. Specifically, the Qwen2VL series (i.e., 2B, 7B, and 72B) (Wang et al., 2024a) adopts an end-to-end vision-language architecture; the InternVL2 series (i.e., 1B, 8B, and 26B) (Chen et al., 2024b) emphasizes high-resolution visual encoding and cross-modal alignment; the LLaVA-1.5 series (i.e., 7B and 13B) (Liu et al., 2023) employs a lightweight CLIP-to-LLM projection (Radford et al., 2021); and the LLaVA-NeXT series (i.e., 7B and 13B) (Liu et al., 2024b) further improves resolution and training efficiency.

**Implementation Details.** Following prior works (Chen et al., 2024a; Farquhar et al., 2024), the parameter configuration for all models in the experiments is set to *temperature*

= 0.1 when getting the reference answer, and *temperature* = 0.5, *top-k* = 10, *top-p* = 0.99 when generating responses for uncertainty evaluation. The sampling time $K$ is set to 10. The threshold $\theta$ we use to measure the confidence of responses is 0.48 based on the parameter analysis in Figure 3c.

**Baselines.** We compare our method with six representative baselines, i.e., GAVIE (Liu et al., 2024a) (external LLM-based factuality check), Semantic Entropy (Farquhar et al., 2024) (uncertainty over semantic clusters), KLE (Nikitin et al., 2024) (kernel-based continuous entropy), EigenScore (Chen et al., 2024a) (embedding diversity via covariance eigenvalues), VL-Uncertainty (Zhang et al., 2024) (probability-based uncertainty estimation) and SNNE (Nguyen et al., 2025) (entropy incorporating pairwise semantic similarity). These baselines cover a diverse range of mainstream hallucination detection paradigms, including entropy-based, sampling-based, and metric-based approaches, ensuring a comprehensive and fair evaluation.

## 4.2. Comparison with State-of-the-Art Baselines

Table 1 reports the experimental results across four multimodal benchmarks. From the results, we can see that our DOUBT consistently achieves superior or competitive performance compared to all state-of-the-art baselines.

**Free-form Datasets.** On the LLaVABench dataset, our method achieves an average accuracy of 64.50%, ranking first overall while slightly outperforming EigenScore's 63.83%. Notably, our DOUBT achieves the highest improvement of 8.33% when using the InternVL2-1B model, showing particularly strong adaptation to smaller-scale models. Moreover, gains are more pronounced on L7B than on L13B. We attribute this to larger models possessing stronger inherent reasoning, thus benefiting less from explicit OUB guidance compared to smaller models. Results on the MMVet dataset show our method attaining the highest average accuracy of 75.23%, significantly surpassing EigenScore's 74.27%. The method maintains consistent leadership across the InternVL2 model series, achieving 78.44%, 72.02%, and 75.23% on the I1B, I8B, and I26B models, respectively, demonstrating effective scalability across different model sizes.

**Multiple-choice Datasets.** Our method also exhibits clear advantages on multiple-choice benchmarks. Specifically, on the MMMU dataset, DOUBT achieves a notable average accuracy of 60.22%, exceeding VL-Uncertainty's 57.68% by a considerable margin of 2.54 percentage points. The method reaches its excellent performance of 68.72% using the Qwen2VL-72B model, highlighting its advantage with larger-scale models. On the ScienceQA dataset, DOUBT achieves the highest average accuracy of 76.07%, outperforming VL-Uncertainty's 75.25%. The approach achieves consistently better performance than baseline methods across most models, demonstrating strong capability in scientific question answering tasks.

Overall, our DOUBT achieves the best average performance on all benchmarks. It maintains consistent performance across various model scales, ranging from 1B to 72B parameters. Other baseline methods fail to achieve consistently strong performance across all datasets. While some methods (e.g., EigenScore) may perform well on individual datasets or models, their performance varies across models and scales. In contrast, DOUBT not only surpasses strong baselines with large margins on datasets like LLaVABench and MMVet, but also maintains stable competitiveness on more difficult benchmarks such as MMMU and ScienceQA, demonstrating its superior generalization and robustness. This is attributed to our use of the OUB strategy and vMF-based trustworthiness, which makes the detection results more accurate and stable compared to other methods that rely solely on multiple sampling and entropy-based measures.

The advantage of DOUBT comes from addressing two limitations that commonly affect existing baselines. First, methods such as Semantic Entropy, KLE, EigenScore, VL-Uncertainty, and SNNE mainly rely on the diversity or dispersion of sampled responses. When an MLLM repeatedly generates semantically similar but incorrect answers, these methods may assign a high confidence score to hallucinated outputs, leading to missed detections. In contrast, DOUBT introduces OUB to decouple object recognition from relational reasoning, which encourages the model to generate object-grounded responses and exposes inconsistencies that are hidden in direct sampling. Second, entropy-based baselines often require reliable semantic clustering or sufficient samples, making them unstable in low-sample regimes. The proposed vMF-based trustworthiness directly measures the directional concentration of normalized response embeddings, avoiding discrete clustering and providing a smoother reliability estimate. Compared with external-verification methods such as GAVIE, DOUBT does not rely on additional knowledge bases or white-box model access, making it more generally applicable to black-box MLLMs. These properties explain why DOUBT achieves stronger and more stable performance across different datasets and model families.

## 4.3. Ablation Study

To verify the contribution of each component, we conduct detailed ablation experiments on the larger MMMU and ScienceQA datasets to reduce the randomness caused by limited data size. Table 2 reports the ablation results, where two key components are removed: OUB and vMF-based Trustworthiness. From the results, we observe that removing either component may improve performance in certain test scenarios, but overall performance is negatively affected. For example, after removing OUB, the Qwen2VL-2B model shows improved performance on the MMMU dataset, and removing the vMF-based Trustworthiness also leads to better results on the ScienceQA dataset. However, the full DOUBT method consistently achieves the highest average scores (i.e., 60.22% on MMMU and 76.07% on ScienceQA), surpassing all ablated variants. These results demonstrate that both components are essential and complementary: OUB enriches response diversity and semantic grounding, while vMF-based Trustworthiness provides a stable and geometry-aware measure of reliability.

## 4.4. Parameter Study

We conduct a sensitivity analysis on answer-generation hyperparameters, including temperature, top-$p$, sampling number $K$, and threshold, using Qwen2VL-2B on MMVet, with LLaVA-NeXT-7B additionally evaluated for threshold analysis. As shown in Figure 3, the model exhibits distinct behaviors under different hyperparameter settings. Temper-

| Dataset | Configuration | Q2B | Q7B | Q72B | I1B | I8B | I26B | L7B | L13B | LN7B | LN13B | Avg |
|---|---|---|---|---|---|---|---|---|---|---|---|---|
| MMMU | Full version | 60.85 | **64.61** | **68.72** | 57.58 | 59.39 | 59.64 | 58.55 | 56.48 | **60.85** | 55.52 | **60.22** |
| | - w/o OUB | **61.94** | 62.18 | 66.30 | 55.27 | 59.64 | **59.88** | 59.27 | 57.94 | 59.88 | 55.39 | 59.77 |
| | - w/o vMF | 60.96 | 59.35 | 67.15 | **59.61** | **60.73** | 55.39 | **59.76** | **59.96** | 58.84 | **58.38** | 60.01 |
| | - w/o OUB & vMF | 56.73 | 58.91 | 59.88 | 57.09 | 54.55 | 55.15 | 56.97 | 56.36 | 57.58 | 53.82 | 56.70 |
| ScienceQA | Full version | 65.00 | **79.87** | **88.80** | 68.32 | **90.08** | 91.37 | 66.48 | **72.43** | 70.95 | 67.43 | **76.07** |
| | - w/o OUB | 65.99 | 79.03 | 88.00 | 68.12 | 89.99 | **91.52** | **66.98** | 70.90 | **71.15** | 67.58 | 75.93 |
| | - w/o vMF | **66.21** | 65.47 | 61.63 | 63.99 | 69.33 | 81.11 | 63.91 | 64.71 | 66.04 | **68.57** | 67.10 |
| | - w/o OUB & vMF | 57.26 | 63.66 | 68.82 | 57.11 | 67.38 | 68.66 | 58.55 | 59.74 | 60.29 | 58.80 | 62.03 |

*Table 2.* Ablation study of our method on two datasets by removing two key components. We compare the full method with two variants: without object-level understanding and bridging (- w/o OUB) and without vMF-based trustworthiness (- w/o vMF). The reported numbers are detection accuracies in percentage. **Bold** numbers indicate the best performance.

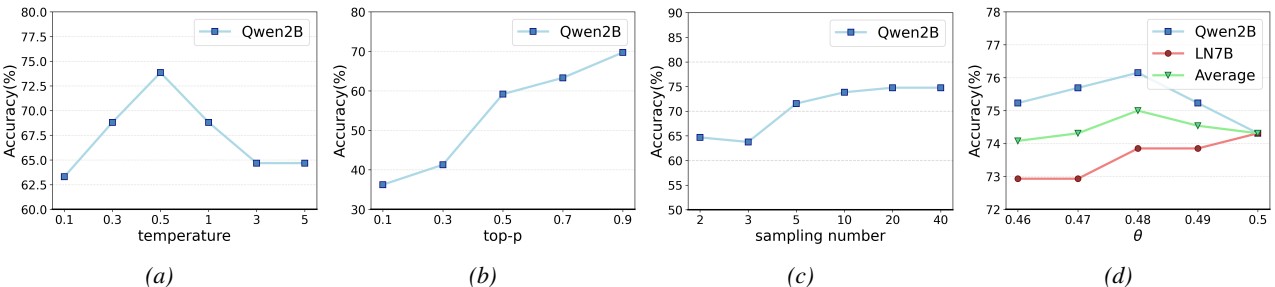

*Figure 3.* (a) Performance sensitivity to temperature. (b) Performance sensitivity to top-p. (c) Performance sensitivity to sampling number. (d) Performance sensitivity to threshold.

ature exhibits a non-monotonic effect, with performance first improving and then saturating, while top-$p$ shows a clear positive correlation with accuracy. For the sampling number $K$, performance improves when $K$ increases from a small value, but gradually saturates after $K = 10$. This suggests that once the main semantic response modes have been covered, additional samples tend to produce repeated or semantically similar outputs, bringing limited new evidence for trustworthiness estimation while increasing inference cost. In contrast, performance is relatively stable across different threshold values for both models, with the best overall performance achieved at a threshold of $0.48$. Overall, top-$p$ has the most significant impact on accuracy, whereas temperature and threshold introduce only moderate variations. We set $K = 10$ as a practical trade-off between detection performance and computational cost.

## 5. Computational Cost Analysis

To further evaluate the efficiency of DOUBT, we compare its computational cost with representative baselines, including VL-Uncertainty and EigenScore. As shown in Table 3, we report the average latency, generated output tokens, and the number of MLLM forward passes on Qwen2VL-2B. Compared with the baselines, DOUBT incurs higher computational cost due to its two-stage design. Specifically, it requires one additional forward pass for object-level under-

*Table 3.* Computational cost comparison on Qwen2VL-2B. We report the average time per sample, average generated output tokens, and the number of MLLM forward passes.

| Dataset | Method | Time (s) | Avg. Tokens | Forward Passes |
|---|---|---|---|---|
| LLaVABench | VL-Uncertainty | 9.82 | 148.43 | 5 |
| | EigenScore | 20.33 | 991.02 | 10 |
| | Ours | 34.01 | 1739.30 | 21 |
| MMVet | VL-Uncertainty | 6.30 | 68.09 | 5 |
| | EigenScore | 8.24 | 293.42 | 10 |
| | Ours | 15.36 | 554.71 | 21 |
| MMMU | VL-Uncertainty | 6.88 | 40.02 | 5 |
| | EigenScore | 5.14 | 154.34 | 10 |
| | Ours | 9.51 | 275.03 | 21 |
| ScienceQA | VL-Uncertainty | 2.56 | 9.74 | 5 |
| | EigenScore | 1.58 | 19.18 | 10 |
| | Ours | 3.56 | 56.64 | 21 |

standing and $2K$ forward passes for sampling the original and object-bridged response sets, resulting in $2K + 1 = 21$ forward passes when $K = 10$. This leads to higher latency and more generated tokens, especially on free-form datasets such as LLaVABench and MMVet. However, the overall cost remains within a practical range, and the improved detection accuracy justifies the additional overhead. Therefore, DOUBT provides a favorable trade-off between detection performance and computational efficiency.

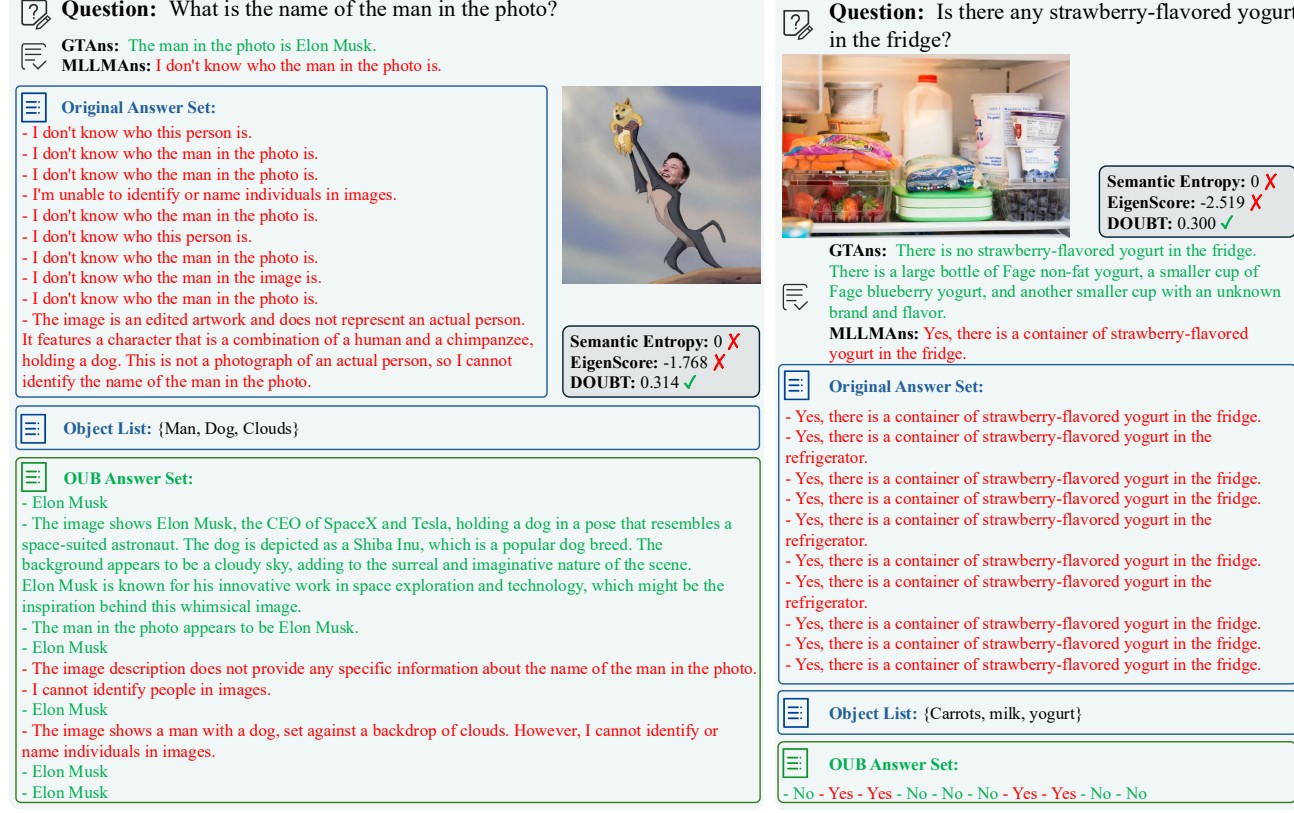

*Figure 4.* Examples of hallucination detection cases. Compared to our method, the other approaches tend to produce falsely consistent results and achieve lower hallucination metrics.

## 5.1. Case Study

We further illustrate the interpretability and reliability of DOUBT with qualitative examples in Figure 4. In both cases, the model initially produces incorrect answers, which can mislead existing detection methods. With DOUBT, the model first performs object-level understanding to identify key entities, and then applies object-level bridging to re-reason the question grounded on these entities. As a result, the generated responses become more consistent with the visual evidence, leading to more reliable trustworthiness estimates. These examples demonstrate how OUB diversifies the response space and refines trustworthiness estimation.

**Hallucination Mitigation.** Beyond detection, DOUBT can also mitigate hallucination through OUB. When the original answer is detected as unreliable, the object-bridged responses can be used as alternative candidates because they are generated by explicitly conditioning reasoning on recognized visual objects. This object-grounded reasoning reduces the model's reliance on language priors and helps produce answers that are more consistent with the image. The qualitative examples in Figure 4 show that OUB can correct confidently hallucinated responses, and more quantitative results are provided in Appendix A.

## 6. Conclusion

In this paper, we investigate hallucination detection in MLLMs, a critical step toward ensuring trustworthy and safe deployment. We propose DOUBT, a black-box reliability assessment framework that integrates step-by-step reasoning prompting and geometric trustworthiness modeling. Specifically, the proposed Object-level Understanding and Bridging (OUB) mechanism decouples visual reasoning from relational reasoning, enabling more diverse and semantically grounded answers. Meanwhile, the vMF-based Trustworthiness metric leverages the mean resultant vector length of the response embeddings to measure trustworthiness in a stable and geometry-aware manner, offering a superior alternative to traditional entropy-based approaches. Extensive experiments across four benchmarks and ten MLLMs demonstrate DOUBT's superior performance and consistent generalization across model scales from 1B to 72B parameters. Unlike previous methods that rely solely on sampling or entropy measures, our DOUBT provides a theoretically grounded and practically robust approach to hallucination detection. We hope our findings inspire future research toward interpretable, uncertainty-aware multimodal reasoning and contribute to the development of more reliable MLLMs.

## Impact Statement

This work aims to improve the reliability of MLLMs by detecting hallucinated outputs. By providing trustworthiness estimates rather than directly modifying model behavior, our approach can support safer deployment of MLLMs in downstream applications. While the method may be used in high-stakes settings, it does not automate decision-making and is intended to complement, rather than replace, human judgment.

## Acknowledgments

This work was supported in part by the National Natural Science Foundation of China under Grant U25B6003, 62472295, and U25A20534; in part by Foundation Enhancement Program Project (Technology Field Fund) under Grant 2025-JCJQ-JJ-0686; in part by Sichuan Science and Technology Planning Project under Grant 24NSFTD0130; in part by the Luzhou City School-Local-Enterprise-Academy Science and Technology Cooperation Project under Grant 2024XDY200; and in part by the Fundamental Research Funds for the Central Universities under Grant CJ202303 and CJ202403.

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

## A. Hallucination Mitigation through OUB

| Dataset | Configuration | Q2B | Q7B | Q72B | I1B | I8B | I26B | L7B | L13B | LN7B | LN13B | Avg |
|---|---|---|---|---|---|---|---|---|---|---|---|---|
| MME | Initial | 82.22 | **88.33** | **91.53** | 54.17 | 71.02 | **79.91** | 76.71 | **74.77** | 58.00 | **73.63** | 75.03 |
| | With OUB | **82.31** | **88.33** | **91.53** | **54.21** | **72.24** | 79.87 | **76.79** | 74.43 | **62.55** | 73.17 | **75.54** |
| POPE (random) | Initial | 89.77 | **88.63** | **89.70** | 67.47 | 80.63 | 72.57 | 75.60 | 87.63 | 56.40 | 87.13 | 79.55 |
| | With OUB | **89.90** | 88.60 | 89.60 | **67.90** | **81.40** | **73.13** | **77.37** | **88.67** | **59.33** | **88.27** | **80.42** |
| POPE (adversarial) | Initial | **87.37** | **86.20** | **86.37** | **65.13** | 77.47 | 69.07 | 71.87 | 79.80 | 49.27 | **83.40** | 75.60 |
| | With OUB | 87.30 | 86.13 | 86.33 | 64.33 | **78.27** | **70.10** | **72.57** | **80.3** | **51.17** | 82.17 | **75.87** |
| POPE (popular) | Initial | 88.37 | **87.50** | **88.33** | 66.53 | 79.07 | 70.87 | 74.73 | 84.70 | 49.30 | **86.00** | 77.54 |
| | With OUB | **88.53** | 87.47 | 88.20 | **66.60** | **79.50** | **71.50** | **75.57** | **85.23** | **51.33** | 85.73 | **77.97** |

*Table 4.* Performance of hallucination mitigation achieved through OUB. The results are reported as answer accuracy in percentage. **Bold** numbers indicate the best performance.

To examine the effectiveness of OUB in correcting hallucinated responses, we conduct experiments on the POPE (Li et al., 2023b) and MME (Fu et al., 2025) datasets. We first prompt the model to generate multiple responses and then use the vMF-based trustworthiness metric to decide whether to apply OUB for hallucination mitigation. When necessary, OUB guides the model to generate a caption for the image, identify the objects mentioned in the caption, and finally produce a refined answer based on both the caption and the object list. Table 4 presents the answer accuracy of different models before and after applying OUB. The results show that with OUB, models achieve higher accuracy in most cases, indicating that OUB effectively mitigates hallucination. Therefore, we incorporate OUB into DOUBT to expand the diversity of the model's answer space, making uncertainty detection more reliable and consequently improving hallucination detection performance.

## B. Performance under Different vMF Parameters

| Dataset | Configuration | Q2B | Q7B | Q72B | I1B | I8B | I26B | L7B | L13B | LN7B | LN13B | Avg |
|---|---|---|---|---|---|---|---|---|---|---|---|---|
| MMVet | Ours | **73.85** | **72.94** | **71.10** | **78.44** | 72.02 | 75.23 | **76.61** | **77.98** | **76.15** | **77.98** | **75.23** |
| | Ours ($\kappa$) | 72.48 | 72.48 | 70.18 | 77.98 | **72.48** | **77.06** | **76.61** | 77.52 | **76.15** | 77.52 | 75.05 |
| MMMU | Ours | 60.84 | **64.61** | **68.72** | **57.58** | **59.39** | 59.64 | **58.55** | 56.48 | 60.85 | **55.52** | **60.22** |
| | Ours ($\kappa$) | **60.97** | 63.88 | **68.72** | 55.88 | 58.91 | **59.88** | 57.21 | 55.76 | **60.97** | 55.03 | 59.72 |

*Table 5.* Ablation study of our method on MMVet and MMMU datasets under different vMF parameter settings. Results are reported for both $\bar{R}$ and $\kappa$, with detection accuracies given in percentage. **Bold** numbers indicate the best performance.

Table 5 presents the results obtained using different parameters in vMF, where after normalizing $\kappa$ we set the threshold to 0.38. It can be observed that the overall performance gap between using $\bar{R}$ and $\kappa$ is not substantial, especially in terms of the average scores across datasets. This observation is consistent with the mathematical relationship discussed earlier between the two parameters. However, in terms of overall performance, using $\kappa$ is still somewhat worse than using $\bar{R}$, which further confirms our earlier point that the wide range of $\kappa$ makes it challenging to determine an effective threshold.

## C. Effect of Introducing a More Capable External Model

For a further investigation of the effect of OUB, we replace the model's own object list with that provided by the Qwen-VL-Plus model (Bai et al., 2023). Table 6 presents a comparison of hallucination detection performance before and after applying OUB with a more capable external model. It can be observed that in most cases, the detection performance improves, indicating that when the model performs OUB using more accurate information, it is better able to produce correct answers and the results align with our expectations.

| Dataset | Configuration | Q2B | I8B | L13B | LN7B | Avg |
|---|---|---|---|---|---|---|
| LLaVABench | Ours | 68.33 | 61.67 | 66.67 | 63.33 | 65.00 |
| | With QVP | **71.67** | **63.33** | **70.00** | **65.00** | **67.50** |
| MMVet | Ours | **73.85** | 72.02 | 77.98 | **76.15** | 75.00 |
| | With QVP | 72.02 | **74.77** | **81.19** | **76.15** | **76.03** |
| ScienceQA | Ours | 65.00 | 90.08 | **72.43** | 70.95 | 74.62 |
| | With QVP | **66.39** | **90.38** | 71.19 | **72.68** | **75.16** |

*Table 6.* Ablation study on the effect of introducing a more capable model. QVP denotes Qwen-VL-Plus. Results are reported as detection accuracies in percentage. **Bold** numbers indicate the best performance.

## D. Impact of Different Uncertainty Estimation Metrics

| Dataset | Metric | Q2B | Q7B | Q72B | I1B | I8B | I26B | L7B | L13B | LN7B | LN13B | Avg |
|---|---|---|---|---|---|---|---|---|---|---|---|---|
| LLaVABench | SE | **68.33** | **58.33** | 55.00 | 71.67 | **63.33** | 53.33 | **80.00** | 63.33 | **65.00** | 63.33 | 64.17 |
| | ES | 48.33 | **58.33** | **60.00** | 50.00 | 60.00 | **61.67** | 50.00 | **66.67** | 60.00 | 61.67 | 57.67 |
| | vMF-T | **68.33** | **58.33** | 53.33 | **73.33** | 61.67 | 55.00 | **80.00** | **66.67** | 63.33 | **65.00** | **64.50** |
| MMVet | SE | 68.46 | 70.85 | 59.17 | 71.25 | 71.23 | 69.50 | 69.91 | 71.67 | 73.85 | 69.57 | 69.55 |
| | ES | 69.72 | **76.61** | **72.02** | 75.69 | **73.85** | **77.06** | 76.15 | **80.28** | 73.31 | 77.06 | 75.18 |
| | vMF-T | **73.85** | 72.94 | 71.10 | **78.44** | 72.02 | 75.23 | **76.61** | 77.98 | **76.15** | **77.98** | **75.23** |
| MMMU | SE | **60.96** | 59.35 | 67.15 | **59.61** | 60.73 | 55.39 | **59.76** | 59.96 | 58.84 | **58.38** | 60.01 |
| | ES | 54.06 | **66.55** | 68.85 | 53.58 | **61.09** | 55.52 | 56.48 | 56.24 | 56.61 | 50.67 | 57.97 |
| | vMF-T | 60.85 | 64.61 | **68.72** | 57.58 | 59.39 | **59.64** | 58.55 | 56.48 | **60.85** | 55.52 | **60.22** |
| ScienceQA | SE | 66.21 | 65.47 | 61.63 | 63.99 | 69.33 | 81.11 | 63.91 | 64.71 | 66.04 | **68.57** | 67.10 |
| | ES | 61.87 | 72.24 | 85.57 | 56.37 | 59.74 | 85.72 | 63.11 | 68.77 | 70.90 | 63.41 | 68.77 |
| | vMF-T | 65.00 | **79.87** | **88.80** | **68.32** | **90.08** | **91.37** | **66.48** | **72.43** | **70.95** | 67.43 | **76.07** |

*Table 7.* Comparison of different uncertainty estimation metrics. SE, ES, and vMF-T denote Semantic Entropy, EigenScore and vMF-based Trustworthiness. Results are reported as detection accuracies in percentage. **Bold** numbers indicate the best performance.

When calculating uncertainty, the specific metric used has a significant impact on the final results. To compare the performance of different metrics, we apply each metric to DOUBT and examine the outcomes. Table 7 presents the results of using different uncertainty estimation metrics in DOUBT. It can be clearly observed that when using our proposed vMF-based trustworthiness metric, the overall performance remains stable, and it achieves the highest average score across all benchmarks. This demonstrates the superiority of the vMF-based trustworthiness metric.

## E. Ablation on the Number of Recognized Objects

In Section 3.2, we limit the object-level understanding prompt to extract up to three main objects. Here, we further analyze whether the number of recognized objects affects the performance of DOUBT. It is worth noting that the number of sampling iterations $K$ is independent of the number of recognized objects $N$. The sampling number $K$ controls how many complete responses are generated for trustworthiness estimation, while $N$ only controls the maximum number of objects included in the object list. Therefore, DOUBT does not generate one answer per object; instead, the recognized object list is used as a shared object-level context for generating each object-bridged response.

Table 8 reports the ablation results by varying the maximum number of recognized objects from $N = 1$ to $N = 10$, together with an adaptive "All objects" setting. The results show that increasing the number of objects does not consistently improve

*Table 8.* Ablation on the maximum number of recognized objects $N$ in OUB. Results are reported as detection accuracies in percentage. **Bold** numbers indicate the best performance.

| MMVet | Qwen2VL-2B | InternVL2-1B |
|---|---|---|
| 1 obj | 72.02 | 78.44 |
| 2 obj | 68.35 | 76.61 |
| 3 obj | **73.85** | 78.44 |
| 4 obj | 71.10 | 78.44 |
| 5 obj | 69.72 | **79.82** |
| 10 obj | 68.35 | 75.23 |
| All objects | 71.10 | 77.52 |

| ScienceQA | Qwen2VL-2B | InternVL2-1B |
|---|---|---|
| 1 obj | 66.63 | 66.98 |
| 2 obj | 65.89 | 67.13 |
| 3 obj | 65.00 | **68.32** |
| 4 obj | **67.38** | 68.12 |
| 5 obj | 66.44 | 67.63 |
| 10 obj | 66.93 | 66.88 |
| All objects | 66.24 | 66.88 |

performance. On MMVet, Qwen2VL-2B achieves the best result at $N = 3$, while InternVL2-1B performs best at $N = 5$. On ScienceQA, the best results are obtained at $N = 4$ for Qwen2VL-2B and $N = 3$ for InternVL2-1B. In contrast, using too many objects, such as $N = 10$ or all recognized objects, often leads to degraded performance. This suggests that including more objects may introduce long-tail noise or irrelevant distractors, which can weaken the grounding effect of OUB.

These results validate our top-3 object selection heuristic as a robust and cost-effective choice. Although images may contain many objects, not all of them are equally relevant to the target question. Selecting the most salient objects helps preserve the main visual context while avoiding excessive distractors. Therefore, the top-3 setting does not necessarily cause context-missing hallucinations; instead, it provides a practical balance between scene coverage and noise suppression.

## F. Hard Case Analysis

To further illustrate the advantage of DOUBT in handling falsely consistent hallucinations, we conduct a hard-case analysis on ScienceQA. We define hard cases as samples where the reference answer is incorrect while Semantic Entropy (SE) is equal to 0. These cases indicate a typical false-consistency failure mode: the model repeatedly generates semantically similar but incorrect responses, causing uncertainty-based methods such as SE to regard them as reliable. Therefore, by definition, SE fails to detect hallucinations on this subset.

Table 9 reports the number of hard cases and the number of cases successfully detected by DOUBT. Across different MLLMs, DOUBT consistently recovers a substantial portion of these falsely consistent hallucinations. For example, DOUBT successfully detects 67 out of 165 hard cases for Qwen2VL-7B and 88 out of 233 hard cases for InternVL2-1B. Overall, DOUBT detects 393 out of 1005 hard cases, achieving a recovery rate of 39.10%. These results demonstrate that DOUBT can effectively break the false consistency of hallucinated responses by introducing object-level understanding and bridging, thereby revealing unreliable predictions that are difficult to identify from direct response sampling alone.

## G. Robustness to Stage-2 Prompt Formats

DOUBT uses a fixed Stage-2 bridging prompt in the main experiments. To examine whether the effectiveness of DOUBT depends on a carefully selected prompt format, we further evaluate several alternative Stage-2 prompts, including concise instruction prompts, zero-shot Chain-of-Thought (CoT) prompts, and few-shot CoT prompts. All variants follow the same

*Table 9.* Hard-case analysis on ScienceQA. Hard cases are defined as samples where the reference answer is incorrect while Semantic Entropy is equal to 0. Recovered cases denote hard cases successfully detected by DOUBT.

| Model | Hard Cases | Recovered Cases | Recovery Rate |
|---|---|---|---|
| Qwen2VL-2B | 121 | 50 | 41.32% |
| Qwen2VL-7B | 165 | 67 | 40.61% |
| Qwen2VL-72B | 120 | 51 | 42.50% |
| InternVL2-1B | 233 | 88 | 37.77% |
| LLaVA-1.5-7B | 207 | 73 | 35.27% |
| LLaVA-NeXT-7B | 159 | 64 | 40.25% |
| Total | 1005 | 393 | 39.10% |

*Table 10.* Robustness analysis of DOUBT under different Stage-2 prompt formats. Results are reported as detection accuracies in percentage. **Bold** numbers indicate the best performance.

| LLaVABench | Qwen2VL-2B | InternVL2-1B |
|---|---|---|
| Instruction | 70.00 | 66.67 |
| CoT (0-shot) | 70.00 | 70.00 |
| CoT (few-shot) | **73.33** | 61.67 |
| Ours | 68.33 | **73.33** |
| **MMVet** | **Qwen2VL-2B** | **InternVL2-1B** |
| Instruction | 71.10 | 67.89 |
| CoT (0-shot) | 68.81 | 75.69 |
| CoT (few-shot) | **74.77** | 69.72 |
| Ours | 73.85 | **78.44** |

overall DOUBT pipeline and differ only in the format of the Stage-2 bridging prompt.

Table 10 reports the results on LLaVABench and MMVet with Qwen2VL-2B and InternVL2-1B. The results show that DOUBT remains effective across different prompt formats, indicating that its performance does not rely on a single manually tuned prompt. Moreover, the best prompt format varies across models and datasets. For example, CoT with few-shot examples achieves the best performance for Qwen2VL-2B on both LLaVABench and MMVet, while the original prompt performs best for InternVL2-1B. These results suggest that the default Stage-2 prompt used in the main experiments is a strong general choice, while DOUBT can also flexibly benefit from other prompt styles depending on the underlying MLLM.

## H. Robustness to Different Semantic Embedding Models

In the main experiments, we use `nli-roberta-large` as the semantic embedding model for fair comparison with prior embedding-based methods such as EigenScore. To evaluate the robustness of DOUBT to different embedding encoders, we replace `nli-roberta-large` with `Qwen3-Embedding-0.6B` (Zhang et al., 2025b), `all-MiniLM-L6-v2` (Reimers & Gurevych, 2019; Wang et al., 2020), and `bge-base-en-v1.5` (Xiao et al., 2023). All other settings remain unchanged.

As shown in Table 11, DOUBT achieves stable performance across different semantic encoders. On LLaVABench, the average accuracy varies only from 63.33% to 64.50%, and the lightweight `all-MiniLM-L6-v2` model achieves 64.00%, close to the result of `nli-roberta-large`. On MMMU, the average accuracy remains within a narrow range from 59.94% to 60.22%. These results indicate that the proposed vMF-based trustworthiness metric is not tightly coupled with a specific embedding model and can maintain stable performance even when smaller encoders are used.

*Table 11.* Robustness analysis of DOUBT with different semantic embedding models. Results are reported as detection accuracies in percentage.

| LLaVABench | Q2B | Q7B | Q72B | I1B | I8B | I26B | L7B | L13B | LN7B | LN13B | Avg |
|---|---|---|---|---|---|---|---|---|---|---|---|
| Qwen3-Embedding-0.6B | 66.67 | 56.67 | 51.67 | 71.67 | 60.00 | 53.33 | 80.00 | 66.67 | 63.33 | 63.33 | 63.33 |
| all-MiniLM-L6-v2 | 66.67 | 56.67 | 53.33 | 71.67 | 61.67 | 55.00 | 80.00 | 66.67 | 63.33 | 65.00 | 64.00 |
| bge-base-en-v1.5 | 66.67 | 56.67 | 53.33 | 71.67 | 61.67 | 53.33 | 80.00 | 66.67 | 63.33 | 65.00 | 63.83 |
| Ours (nli-roberta-large) | 68.33 | 58.33 | 53.33 | 73.33 | 61.67 | 55.00 | 80.00 | 66.67 | 63.33 | 65.00 | 64.50 |

| MMMU | Q2B | Q7B | Q72B | I1B | I8B | I26B | L7B | L13B | LN7B | LN13B | Avg |
|---|---|---|---|---|---|---|---|---|---|---|---|
| Qwen3-Embedding-0.6B | 60.48 | 63.88 | 68.00 | 57.70 | 59.39 | 59.15 | 58.30 | 56.24 | 61.09 | 55.15 | 59.94 |
| all-MiniLM-L6-v2 | 59.64 | 64.61 | 68.12 | 58.18 | 59.03 | 59.39 | 58.30 | 55.88 | 61.45 | 56.36 | 60.10 |
| bge-base-en-v1.5 | 59.88 | 64.48 | 68.24 | 57.82 | 59.27 | 59.39 | 58.18 | 56.61 | 61.21 | 55.76 | 60.08 |
| Ours (nli-roberta-large) | 60.84 | 64.61 | 68.72 | 57.58 | 59.39 | 59.64 | 58.55 | 56.48 | 60.85 | 55.52 | 60.22 |

# I. Case Study

We have selected more cases for illustration to better demonstrate the effectiveness of DOUBT. Their detection processes are consistent with our initial idea of increasing the diversity of the answer space. Even though a few cases may not show this effect clearly, they can still be detected through the more stable vMF-based trustworthiness metric.

**Question:** How many uncut fruits are in the image?

**GTAns:** There are three uncut fruits in the image.
**MLLMAns:** There are four uncut fruits in the image.

**Original Answer Set:**
- *There are three uncut fruits in the image.*
- *There are four uncut fruits in the image.*
- *There are four uncut fruits in the image.*
- *There are three uncut fruits in the image.*
- *There are four uncut fruits in the image.*
- *There are four uncut fruits in the image.*
- *There are three uncut fruits in the image.*
- *There are four uncut fruits in the image.*
- *There are three uncut fruits in the image.*
- *There are four uncut fruits in the image.*

**Object List:** {Fruit, Fruit, Fruit}

**OUB Answer Set:**
- *4 - 4 - 5 - 4 - 3 - 4 - 5 - 4 - 2 - 4*

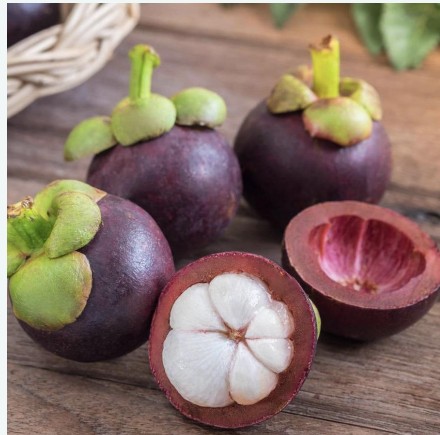

**Semantic Entropy:** 0.971 ✗
**EigenScore:** -2.325 ✗
**DOUBT:** 0.269 ✓

---

**Question:** What is the total price for a bottle of Merlot and a bottle of Cabernet shown in the image?

**GTAns:** 249.98
**MLLMAns:** 248.98

**Original Answer Set:**
- *249.98 - $249.98 - 249.98 - 248.98 - 248.98 - 248.98 - 248.98 - 249.98 - 248.98 - 249.98*

**Object List:** {sign, wine bottle, person}

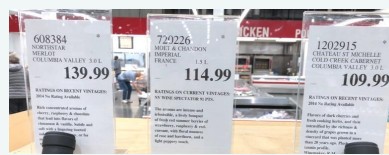

**Semantic Entropy:** 1 ✗
**EigenScore:** -2.155 ✗
**DOUBT:** 0.275 ✓

**OUB Answer Set:**
- *The total price for a bottle of Merlot and a bottle of Cabernet is $249.98.*
- *The total price for a bottle of Merlot and a bottle of Cabernet shown in the image is $248.98.*
- *$249.98*
- *253.98*
- *The total price for a bottle of Merlot and a bottle of Cabernet shown in the image is 248.98.*
- *The total price for a bottle of Merlot (139.99) and a bottle of Cabernet (109.99) is 249.98.*
- *The total price for a bottle of Merlot and a bottle of Cabernet shown in the image is 248.98.*
- *The total price for a bottle of Merlot and a bottle of Cabernet shown in the image is $249.98.*
- *249.98*
- *The total price for a bottle of Merlot and a bottle of Cabernet shown in the image is 248.98.*

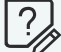 **Question:** Are which one of these two jets are larger in size?

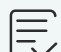 **GTAns:** same.
**MLLMAns:** The jet on the left is larger in size.

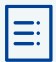 **Original Answer Set:**

- *The jet in the foreground is larger in size.*
- *The jet on the left is larger in size.*
- *The jet on the left is larger in size.*
- *The jet on the right is larger in size.*
- *The jet on the left is larger in size.*
- *The jet in the foreground is larger in size.*
- *The jet in the foreground is larger in size.*
- *The jet on the left is larger in size.*
- *The jet on the left is larger in size.*
- *The jet on the left is larger in size.*

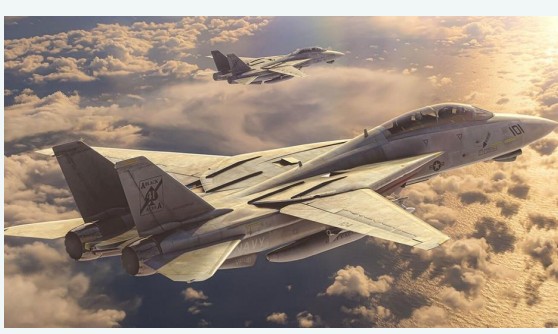

**Semantic Entropy:** 1.295 ✓
**EigenScore:** -2.152 ✗
**DOUBT:** 0.231 ✓

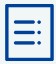 **Object List:** {Fighter jet, clouds, ocean}

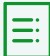 **OUB Answer Set:**

- *The fighter jet in the foreground appears to be larger than the one in the background.*
- *The jet in the foreground is larger in size.*
- *The jet in the foreground appears to be larger than the one in the background.*
- *The jet in the foreground appears to be larger in size compared to the one in the background.*
- *The first jet is larger in size.*
- *The jet in the foreground is larger in size.*
- *The jet in the foreground is larger in size.*
- *The fighter jet in the foreground is larger in size compared to the one in the background.*
- *The jet on the left is larger in size.*
- *The jet in the foreground is larger in size.*

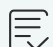 **Question:** Is the surfer to the left or to the right of the surfboard?

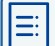 **GTAns:** left
**MLLMAns:** The surfer is to the right of the surfboard

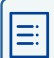 **Original Answer Set:**
- *The surfer is to the right of the surfboard.*
- *The surfer is to the left of the surfboard.*
- *The surfer is to the right of the surfboard.*
- *The surfer is to the right of the surfboard.*
- *The surfer is to the left of the surfboard.*
- *The surfer is to the left of the surfboard.*
- *The surfer is to the right of the surfboard.*
- *The surfer is to the right of the surfboard.*
- *To the left.*
- *The surfer is to the left of the surfboard.*

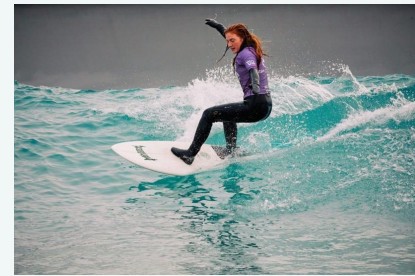

**Semantic Entropy:** 1.361 ✓
**EigenScore:** -2.045 ✗
**DOUBT:** 0.164 ✓

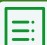 **Object List:** {surfer, surfboard, ocean}

**OUB Answer Set:**
- *The surfer is to the left of the surfboard.*
- *The surfer is to the right of the surfboard.*
- *To the right*
- *The surfer is to the left of the surfboard.*
- *To determine if the surfer is to the left or right of the surfboard, let's analyze the image description provided:\n\n1. **Identify the Surfer**: The image shows a surfer who is in the air, performing a maneuver.\n2. **Identify the Surfboard**: The surfboard is clearly visible and the surfer is standing on it.\n3. **Relative Position**: The surfer is positioned on the surfboard, which means they are directly on top of it.\n\nBased on the image description, the surfer is on top of the surfboard, so they are neither to the left*
- *The surfer is to the left of the surfboard.*
- *The surfer is to the right of the surfboard.*
- *The surfer is to the right of the surfboard.*
- *The surfer is to the left of the surfboard.*
- *The surfer is to the left of the surfboard.*

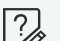 **Question:** Which continent is highlighted?

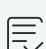 **GTAns:** Asia
**MLLMAns:** Europe

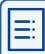 **Original Answer Set:**
- *Asia* - *Asia* - *Europe* - *Europe* - *Europe*
- *Asia* - *Europe* - *Europe* - *Europe* - *Europe*

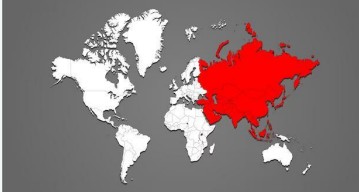

**Semantic Entropy:** 0.881 ✗
**EigenScore:** -2.306 ✗
**DOUBT:** 0.304 ✓

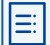 **Object List:** {Europe, Asia, North America}

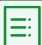 **OUB Answer Set:**
- *Europe and Asia* - *Europe* - *Europe* - *Europe and Asia* - *Europe* - *Europe* - *Europe and Asia* - *Asia* - *Europe and Asia* - *Asia*

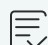 **Question:** Is the person bigger than the car?

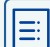 **GTAns:** no
**MLLMAns:** Yes.

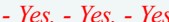 **Original Answer Set:**
- *Yes.* - *Yes.* - *Yes.*
- *Yes, the person is bigger than the car.*
- *no* - *yes* - *yes* - *yes* - *Yes.* - *Yes.*

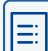 **Object List:** {man, bicycle, car}

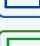 **OUB Answer Set:**
- *Yes, the person is bigger than the car.*
- *Yes*
- *Yes*
- *Yes, the person is bigger than the car.*
- *Yes*
- *Yes, the person is bigger than the car.*
- *No, the person is not bigger than the car.*
- *Yes, the person is bigger than the car.*
- *Yes, the person is bigger than the car.*
- *Yes, the person is bigger than the car.*

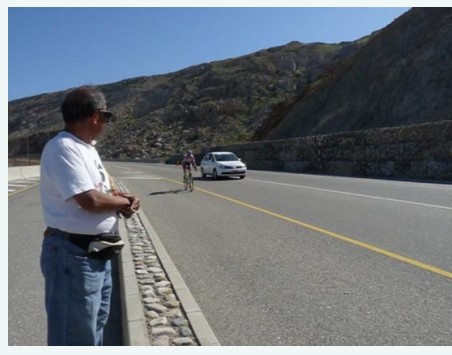

**Semantic Entropy:** 0.922 ✗
**EigenScore:** -1.956 ✓
**DOUBT:** 0.465 ✓

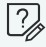 **Question:** Which chest xray disease can be found?
(Options include: Atelectasis; Cardiomegaly; Effusion;
Infiltration; Mass; Nodule; Pneumonia; Pneumothorax;
Consolidation; Edema; Emphysema; Fibrosis;
Pleural Thickening; Hernia.)

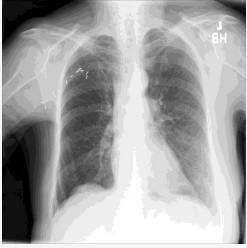

**Semantic Entropy:** 0.881 ✗
**EigenScore:** -2.319 ✗
**DOUBT:** 0.300 ✓

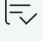 **GTAns:** Hernia
**MLLMAns:** Atelectasis

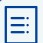 **Original Answer Set:**
- *Cardiomegaly* - *Atelectasis* - *Atelectasis* - *Cardiomegaly* - *Atelectasis* - *Atelectasis* - *Atelectasis* - *Atelectasis* - *Cardiomegaly* - *Atelectasis*

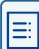 **Object List:** {Lung, heart, ribs}

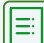 **OUB Answer Set:**
- *Cardiomegaly* - *Cardiomegaly* - *Atelectasis* - *Atelectasis* - *Atelectasis* - *Atelectasis* - *Cardiomegaly* - *Atelectasis* - *Atelectasis* - *Cardiomegaly*

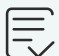 **Question:** How many tomatoes are there?

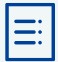 **GTAns:** 5
**MLLMAns:** There are 6 tomatoes.

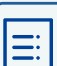 **Original Answer Set:**
- *There are six tomatoes.*
- *There are six tomatoes.*
- *There are 6 tomatoes.*
- *There are 6 tomatoes.*
- *There are six tomatoes.*
- *There are six tomatoes.*
- *There are 8 tomatoes.*
- *There are 7 tomatoes.*
- *There are six tomatoes.*
- *There are 6 tomatoes.*

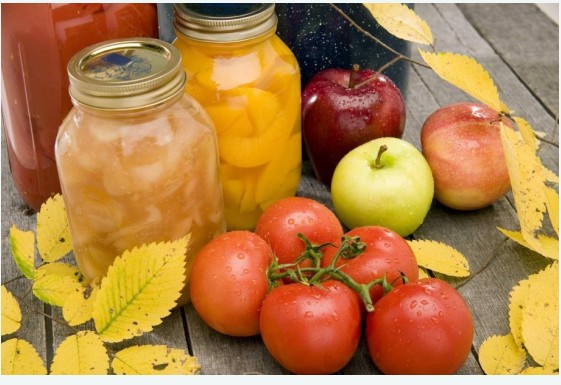

**Semantic Entropy:** 0.922 ✗
**EigenScore:** -2.039 ✓
**DOUBT:** 0.458 ✓

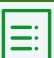 **Object List:** {apple, tomato, jar}

**OUB Answer Set:**
- *There are 6 tomatoes in the image.*
- *There are 7 tomatoes in the image.*
- *There are 7 tomatoes in the image.*
- *There are 9 tomatoes in the image.*
- *There are 6 tomatoes in the image.*
- *There are a total of 7 tomatoes in the image.*
- *There are 6 tomatoes in the image.*
- *There are six tomatoes in the image.*
- *There are 6 tomatoes in the image.*
- *There are six tomatoes in the image.*

**Question:** What is the name of this dish?

**GTAns:** Caprese Salad
**MLLMAns:** Caprese Salad

**Original Answer Set:**
- *Caprese Salad* - *Caprese Salad Skewers* - *Caprese Salad*
- *Caprese Salad* - *Caprese* - *Caprese Salad.* - *Caprese skewers.*
- *Caprese Salad* - *Caprese Salad* - *Caprese Salad*

**Object List:** {Tomato, Mozzarella, Plate}

**OUB Answer Set:**
- *Caprese Salad* - *Caprese Salad* - *Caprese Salad*
- *Caprese Salad* - *Caprese Salad* - *Caprese Salad*
- *Caprese Salad* - *Caprese* - *Caprese Salad*
- *Caprese Salad*

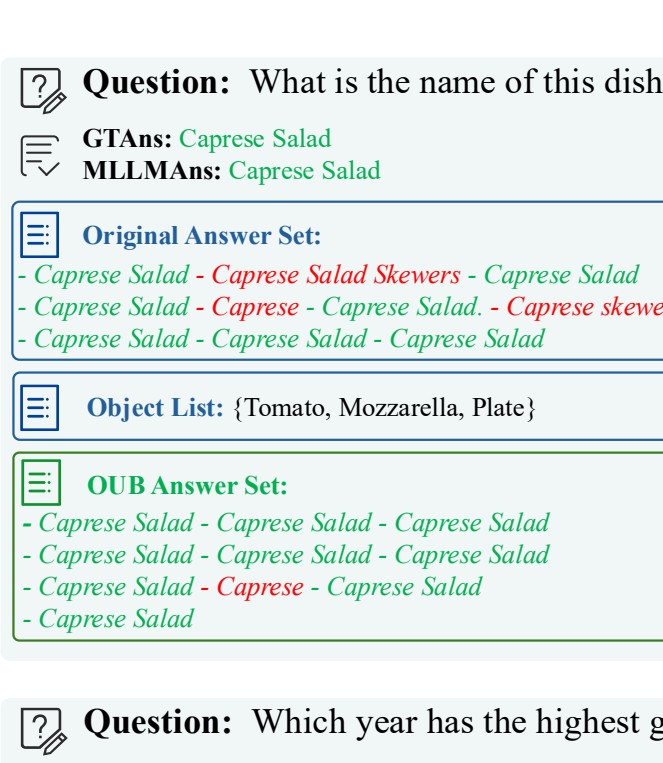

**Semantic Entropy:** 1.157 ✓
**EigenScore:** -1.848 ✗
**DOUBT:** 0.663 ✓

**Question:** Which year has the highest growth rate of median house price?

**GTAns:** 2008
**MLLMAns:** 2008

**Original Answer Set:**
- *2007* - *2008* - *2008* - *2008* - *2008*
- *2008* - *2009* - *2008* - *2008* - *2008*

**Object List:** {Median House Price, Median Gross Rent per Month, Median Household Income}

**OUB Answer Set:**
- *2008* - *2008* - *2008* - *2008* - *2008*
- *2008* - *2008* - *2008* - *2008* - *2008*

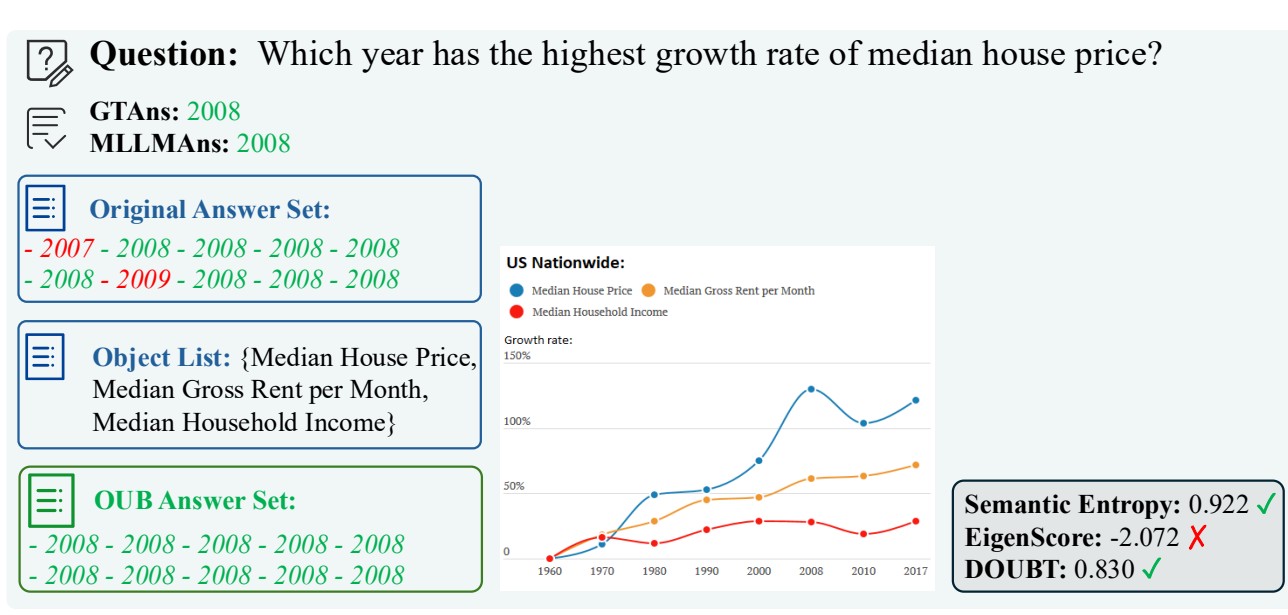

**Semantic Entropy:** 0.922 ✓
**EigenScore:** -2.072 ✗
**DOUBT:** 0.830 ✓

