# OpenReview forum: "DOUBT: Decoupled Object-level Understanding and Bridging via vMF-based Trustworthiness for Hallucination Detection in MLLMs"
_ICML.cc/2026/Conference — ICML 2026 spotlight_

### Official Review · Reviewer_TNUc · 2026-02-20

**Soundness:** 4
**Presentation:** 4
**Significance:** 4
**Originality:** 4
**Overall Recommendation:** 6
**Confidence:** 4

**Summary:**

This paper investigates the problem of hallucination detection in Multimodal Large Language Models (MLLMs). The authors point out that existing consistency-based detection methods are prone to generating "erroneous but consistent" outputs when facing insufficient understanding capabilities of the visual module, leading to missed detections. To address this, the paper proposes the DOUBT framework, which decouples the recognition and reasoning processes through "Object Understanding and Bridging (OUB)" and introduces a geometric confidence metric based on the von Mises-Fisher (vMF) distribution to replace traditional semantic entropy.

**Compliance With Llm Reviewing Policy:**

Affirmed.

**Final Justification:**

My concerns have been resolved, and I am willing to keep my score.

**Key Questions For Authors:**

- The OUB prompt asks for "up to 3 main objects". Was the choice of the number 3 explored systematically?

- Could a dynamic or adaptive number of objects (e.g., based on initial answer complexity) potentially improve results, or is 3 a robust heuristic found empirically?

- The vMF metric uses embeddings from the nli-roberta-largemodel. Was the sensitivity of DOUBT's performance to the choice of this embedding model investigated?      A brief comment on the rationale for this choice or its robustness compared to other sentence encoders would be useful.

**Limitations:**

- The method's effectiveness is inherently tied to the MLLM's underlying object recognition capability. Severely incorrect object lists could propagate errors, though the trustworthiness metric may catch the resulting inconsistency.

- It operates in a black-box setting for hallucination detectionbut not correction. While Appendix A shows OUB can mitigate hallucinations, the core DOUBT framework outputs a binary flag, not a corrected answer

**Strengths And Weaknesses:**

Strengths:
- The authors provide a more stable metric for hallucination detection compared to traditional semantic entropy.
- The proposed DOUBT demonstrates clear advantages on most models and benchmarks.


 Weaknesses:
- The proposed method employs a two-stage prompt strategy, with each stage requiring K sampling iterations, which significantly increases inference latency and API costs. Although the paper briefly mentions cost trade-offs in Section 3.2, it lacks a quantitative efficiency analysis.
- In the object recognition stage, why is the selection limited to "3 main objects"? Could this restriction become a bottleneck for extremely complex image scenes (such as complex charts in ScienceQA)?
- The proposed method relies on semantic embedding models (e.g., nli-roberta-large). Would performance fluctuate significantly if a smaller embedding model were used? Investigating this could help demonstrate stability compared to other methods.
- The proposed method appears to use a fixed prompt format. Was this format meticulously selected? Using different prompts for validation would more fully demonstrate the method's effectiveness.

---

> ### Author Rebuttal · Authors · 2026-03-31
>
> We are extremely grateful for your meticulous evaluation and the highly practical perspective you brought to reviewing our work. We enthusiastically embrace your suggestions, which will undoubtedly make our method more rigorous and generalizable. Please find our detailed responses to your insightful comments below.
>
> > **1. Clarification on Computational Cost and Efficiency (Addressing W1)**
>
> Thank you for highlighting the importance of deployment efficiency. We agree that transparent, quantitative analysis is essential for evaluating practical utility.
>
> To address concerns about sampling and computational overhead, we provide a detailed cost analysis in our response to **Reviewer q6B7**. While our two-stage design introduces additional forward passes (21 in total), the absolute latency remains within a practical range. Given DOUBT’s substantial performance gains, particularly in detecting severe hallucinations missed by baselines, this overhead represents a reasonable trade-off. In the revised Section 3.2, we will add a comprehensive cost table detailing the average token consumption and absolute inference latency to quantitatively demonstrate DOUBT's excellent performance-to-cost ratio.
>
> > **2. Top-3 Object Selection Heuristic and Adaptive Extraction (Addressing W2, Q1, Q2)**
>
> This is an excellent point. We limit the prompt to “up to 3 main objects” to balance attention focus and semantic coverage, as too many objects introduce distracting noise.
>
> To assess whether this limit becomes a bottleneck in complex cases (e.g., ScienceQA), we conduct an ablation over fixed limits ($N=1,2,3,4,5,10$) and an adaptive “All” setting (see **Q2 of Reviewer vdrr**). The results show that the adaptive setting degrades accuracy due to background noise, while performance consistently peaks at $N=3$–$5$. This supports the Top-3 heuristic as a robust and cost-effective choice. We will include this analysis in the revision.
>
> > **3. Sensitivity to the Embedding Model (Addressing W3, Q3)**
>
> Thank you for the insightful comment. We use nli-roberta-large for fair comparison with prior work (e.g., EigenScore).
>
> Our vMF-based metric is designed to be encoder-agnostic. We validate this by replacing it with all-MiniLM-L6-v2, bge-base-en-v1.5, and Qwen3-Embedding-0.6B.
>
> Performance on LLaVABench remains stable across 10 MLLMs, with only 1.17 percentage point variation (63.33%–64.50%). Notably, all-MiniLM-L6-v2 achieves 64.00%, close to nli-roberta-large(64.50%), indicating strong generalizability. Additionally, on the MMMU benchmark, nli-roberta-large achieves an average score of 60.22%, with results across embedding models remaining close to each other, further demonstrating stable performance. We will include this study in the Appendix.
>
> |LLaVABench|Q2B|Q7B|Q72B|I1B|I8B|I26B|L7B|L13B|LN7B|LN13B|Avg|
> |---|---|---|---|---|---|---|---|---|---|---|---|
> | Qwen3-Embedding-0.6B|66.67|56.67|51.67|71.67|60.00|53.33|80.00|66.67|63.33|63.33|63.33|
> | all-MiniLM-L6-v2| 66.67|56.67|53.33|71.67|61.67|55.00|80.00|66.67|63.33|65.00|64.00|
> | bge-base-en-v1.5| 66.67|56.67|53.33|71.67|61.67|53.33|80.00|66.67|63.33|65.00|63.83|
> | Ours (nli-roberta-large) |68.33|58.33|53.33|73.33|61.67|55.00|80.00|66.67|63.33|65.00|64.50|
> | **MMMU**| **Q2B** | **Q7B** | **Q72B** | **I1B** | **I8B** | **I26B** | **L7B** | **L13B** | **LN7B** | **LN13B** | **Avg** |
> | Qwen3-Embedding-0.6B|60.48|63.88|68.00|57.70|59.39|59.15|58.30|56.24|61.09|55.15|59.94|
> | all-MiniLM-L6-v2|59.64|64.61|68.12|58.18|59.03|59.39|58.30|55.88|61.45|56.36|60.10|
> | bge-base-en-v1.5| 59.88|64.48|68.24|57.82|59.27| 59.39|58.18|56.61| 61.21| 55.76     | 60.08   |
> | Ours (nli-roberta-large)| 60.84   | 64.61   | 68.72    | 57.58   | 59.39   | 59.64    | 58.55   | 56.48    | 60.85    | 55.52     | 60.22   |
>
> > **4. Prompt Format Robustness (Addressing W4)**
>
> We initially use a fixed prompt format to control variables, ensuring that performance gains stem from the Object-level Understanding and Bridging (OUB) mechanism rather than prompt design.
>
> We agree that robustness to prompt variation is important. To evaluate this, we conduct an ablation with diverse Stage 2 prompt styles (concise instruction, CoT zero-shot, and CoT few-shot; see **Reviewer vdrr, Q5**).
>
> Results show that OUB remains consistently effective across prompt variations, with certain styles (e.g., CoT few-shot) further improving performance for some models. We will include this study in the Appendix.
>
> > **5. Addressing the Limitations (Addressing L1, L2)**
>
> Thank you for summarizing these important limitations. We acknowledge both issues—error propagation from imperfect object lists (L1) and the distinction between detection and correction (L2).
>
> While our current design partially mitigates these effects (e.g., via inconsistency-based trustworthiness estimation), a more principled treatment remains an important direction. We plan to further explore more robust object extraction and unified detection–correction frameworks in future work.

---

> > ### Author Rebuttal · Reviewer_TNUc · 2026-04-02
> >
> > Thanks for the authors' response. My concerns have been resolved, and I am willing to keep my score.

---

### Official Review · Reviewer_q6B7 · 2026-03-06

**Soundness:** 3
**Presentation:** 3
**Significance:** 3
**Originality:** 3
**Overall Recommendation:** 5
**Confidence:** 4

**Summary:**

This paper proposes a hallucination detection method for MLLM, aiming to address the challenge of existing methods exhibiting highly consistent erroneous answers across multiple forward passes. Specifically, the proposed method includes an object-level understanding and bridging module to induce the model to generate patterns different from those prone to error during direct reasoning, and a vMF-based module to measure consistency based on hallucination results for hallucination detection. Detection results on multiple benchmarks and model families demonstrate the effectiveness and state-of-the-art of the proposed method.

**Compliance With Llm Reviewing Policy:**

Affirmed.

**Final Justification:**

Thanks for the authors' response. My concerns have been resolved, and I am willing to raise my score.

**Key Questions For Authors:**

1. Does the first-stage prompt only require MLLM to recognize 3 objects?
2. Does the first stage run only once per cycle or repeatedly?
3. Why is the semantic entropy -0.0 in Figure 4?

**Limitations:**

No, there is no analysis or discussion about limitations in the text. It is suggested that the author discuss how the limitations in the first phase (when OUB fails) will affect the second phase and other related limitations.

**Strengths And Weaknesses:**

Strengths:

1. This paper addresses an important issue, existing detection methods, such as semantic entropy, fail when faced with multiple similar and erroneous problems.
2. The benchmark and model coverage is comprehensive.
3. Ablation experiments support the proposed module.

Weaknesses:

1. Why does increasing the number of samples not improve performance? This should be discussed in the hyperparameter analysis.
2. A computational cost analysis table should be added to show a comparison with the baseline.
3. The performance in alleviating hallucination should be discussed in the main paper.
4. The baseline methods should be explained and discussed to demonstrate the advantages of DOUBT.
5. The abstract is too long and should be shortened.

---

> ### Author Rebuttal · Authors · 2026-03-31
>
> We deeply appreciate the time and effort you dedicated to reviewing our manuscript. Your constructive suggestions regarding the completeness of our experimental analysis and the structural organization of the paper are invaluable. We have carefully considered all your feedback and formulated a comprehensive revision plan, detailed below.
>
> > #### **Part 1: Enhancements to Manuscript Structure and Experimental Analysis**
>
> We completely agree with your assessment that the manuscript would benefit from a tighter structure and more comprehensive analytical discussions. To address your concerns in the Weaknesses section, we will implement the following systematic updates in the revised manuscript:
>
> 1. **Sampling number analysis.** We will expand the hyperparameter analysis to discuss why increasing $K$ does not improve performance. Specifically, we will explain that performance saturates as the semantic response space becomes bounded, leading to repeated outputs and diminishing returns.
>
> 2. **Computational cost analysis.** We report a detailed computational cost analysis in the table below, including latency, generated tokens, and the number of forward passes. The MLLM we use is Qwen2-VL-2B. DOUBT incurs higher cost primarily due to its two-stage design, requiring one pass for object extraction and $2K$ passes for sampling (21 total when $K=10$). Despite this, the cost remains within a practical range, and the gains in detection accuracy consistently justify the overhead. Overall, DOUBT achieves a favorable and stable performance–efficiency trade-off across benchmarks.
>
> | LLaVABench     | Time per Sample (s)     | Avg Output Tokens     | \# Forward Passes     |
> | -------------- | ----------------------- | --------------------- | --------------------- |
> | VL-Uncertainty | 9.82| 148.43| 5|
> | EigenScore     | 20.33| 991.02| 10|
> | Ours           | 34.01| 1739.30| 21|
> | **MMVet**      | **Time per Sample (s)** | **Avg Output Tokens** | **\# Forward Passes** |
> | VL-Uncertainty | 6.30| 68.09| 5|
> | EigenScore     | 8.24| 293.42| 10|
> | Ours           | 15.36| 554.71 | 21|
> | **MMMU**       | **Time per Sample (s)** | **Avg Output Tokens** | **\# Forward Passes** |
> | VL-Uncertainty | 6.88 | 40.02| 5|
> | EigenScore     | 5.14 | 154.34| 10|
> | Ours           | 9.51| 275.03| 21|
> | **ScienceQA**  | **Time per Sample (s)** | **Avg Output Tokens** | **\# Forward Passes** |
> | VL-Uncertainty | 2.56| 9.74| 5|
> | EigenScore     | 1.58| 19.18| 10|
> | Ours           | 3.56| 56.64| 21|
>
> 3. **Hallucination alleviation.** We will move the analysis of DOUBT’s hallucination mitigation capability from the appendix to the main paper to better highlight its effectiveness.
>
> 4. **Baseline discussion.** We will expand the description and analysis of baseline methods to more clearly demonstrate DOUBT’s advantages.
>
> 5. **Abstract refinement.** We will shorten and refine the abstract to improve clarity and focus.
>
> > #### **Part 2: Clarifications on Key Questions**
>
> > **Q1: Does the first-stage prompt only require MLLM to recognize 3 objects?**
>
> Yes. In our setup, we constrain the prompt to identify up to 3 main objects to balance focused attention and reduce background noise. To validate this choice, we conducted an ablation study varying the number of objects (N=1 to 10, plus an adaptive “All” setting). As detailed in our response to **Q2 of Reviewer vdrr**, we observe that selecting 3–5 objects yields the best performance, while selecting too many objects leads to a drop in performance.
>
> > **Q2: Does the first stage run only once per cycle or repeatedly?**
>
> The Object Recognition prompt (Stage 1) **runs only once per cycle**. Once the object list is generated, it serves as a fixed contextual anchor. Only the bridging reasoning (Stage 2) is sampled $K$ times. Therefore, the total number of MLLM inference calls is $1 + 2K$, rather than $3K$. Because the Stage 1 prompt generates only a very short text (a few tokens), the additional computational overhead is negligible compared to a vanilla sampling baseline that performs $K$ independent reasoning passes, and it does not scale with $K$.
>
> > **Q3: Why is the semantic entropy -0.0 in Figure 4?**
>
> Thank you for pointing this out. The “-0.0” in Figure 4 is a formatting artifact; the true value is exactly 0. We will correct this in the revised figure.
>
> More importantly, this value highlights our core motivation: semantic entropy drops to zero because all $K$ incorrect responses are nearly identical, forming a single cluster with $p=1.0$ (i.e., $-\sum p \log p = 0$). This shows the baseline is misled by consistent but incorrect answers, whereas DOUBT successfully breaks this false consistency. We will clarify this in the figure caption.

---

> > ### Author Rebuttal · Reviewer_q6B7 · 2026-04-04
> >
> > Thanks for the authors' response. My concerns have been resolved, and I am willing to raise my score.

---

### Official Review · Reviewer_vdrr · 2026-03-07

**Soundness:** 3
**Presentation:** 3
**Significance:** 3
**Originality:** 3
**Overall Recommendation:** 4
**Confidence:** 2

**Summary:**

The authors propose a black-box hallucination detection framework named DOUBT. This framework combines an object-aware prompting strategy and a novel confidence scoring method based on the vMF distribution. Experiments cover 10 MLLMs of different architectures and parameter scales, and the method's superior performance over existing baselines is validated on four mainstream benchmark datasets.

**Compliance With Llm Reviewing Policy:**

Affirmed.

**Final Justification:**

After considering both the paper and the authors’ rebuttal, I recommend acceptance. The paper is clear, well motivated, and demonstrates strong empirical performance across a broad range of MLLMs and benchmarks, which makes the work both meaningful and practically relevant. In terms of soundness, the rebuttal addressed my main concerns well, especially by clarifying the fairness of comparisons, explaining the sampling mechanism and top-3 object design, and providing additional evidence on hard cases and prompt robustness. In terms of originality and significance, the proposed object-aware prompting plus vMF-based confidence scoring appears effective and broadly applicable. Overall, the rebuttal strengthened my confidence in the work and positively changed my evaluation.

**Key Questions For Authors:**

Please refer to the Weaknesses section above.

**Limitations:**

Please refer to the Weaknesses section above.

**Strengths And Weaknesses:**

# Strengths
1. The tested models have broad coverage, including mainstream models like LLaVA-1.5, LLaVA-NeXT, Qwen2VL, and InternVL2, ranging from 1B to 72B parameters, which demonstrates the method's general applicability.
2. The proposed method is more stable, maintaining good performance even with fewer sampling iterations.
3. Compared to some classic and newer methods, the proposed method achieves better performance.

# Weaknesses
1. I observe that the experiments compare EigenScore and VL-Uncertainty. Compared to these methods, DOUBT introduces an additional "object recognition" step, which effectively incorporates extra visual priors. If similar "looking at objects before questioning" prompt enhancements were also applied to the baseline methods, would DOUBT's vMF metric still maintain such a significant lead?
2. Is the number of sampling iterations related to the number of recognized objects? One answer per object? If not, would decoupling more objects lead to better results? Furthermore, if an image contains 20 relevant objects, would taking only the top 3 (as mentioned in Section 3.2) cause "context-missing hallucinations"?
3. It is recommended to provide a complete algorithmic pipeline to make the method clearer.
4. It is recommended to provide statistics on hard cases to fully illustrate the method's advantages. For example, the proportion of samples that are consistently wrong would better support the paper's claims.
   Would using different styles of prompts in Stage 2 be more appropriate? This could generate more diverse responses, potentially leading to more accurate vMF score estimation.

---

> ### Author Rebuttal · Authors · 2026-03-31
>
> We sincerely thank you for your insightful and constructive feedback. Your deep understanding of our work and your thoughtful suggestions will significantly improve the quality of our paper. Below, we address your questions and concerns point-by-point.
> > **Q1: Applying OUB to Baselines**
>
> Thank you for this important question on fairness. We agree that isolating the contribution of the vMF metric from the OUB mechanism is essential.
>
> We have conducted this ablation (**Appendix D, Table 6**) by applying OUB prompting to both SE and EigenScore. Even with the same visual priors, vMF-T consistently achieves the best performance across benchmarks (e.g., 76.07% vs. 67.10% for SE and 68.77% for ES on ScienceQA).
>
> This shows that while OUB improves visual grounding, the vMF metric itself is inherently more stable and more effective at capturing semantic consistency in small-sample settings.
>
> > **Q2: Sampling Mechanism & Top-3 Object Selection Heuristic**
>
> Thank you for allowing us to clarify the sampling mechanism.
> 1. **Sampling Mechanism:** The $K$ sampling iterations are not performed per object. Instead, the model first generates a single list of (up to) 3 objects. Then, this combined object list is integrated into a single bridging prompt (Stage 2), and we sample the MLLM $K$ times using temperature scaling based on this unified prompt.
> 2. **Top-3 Object Selection Heuristic:** We limit extraction to up to 3 objects to balance attention focus and scene coverage. To address concerns about complex scenes, we conducted an ablation varying the maximum number of objects ($N=1,2,3,4,5,10$, and an adaptive “All” setting).
>    Results show a clear trend: performance consistently peaks at $N=3$–$5$ across models and datasets (e.g., Qwen2-VL-2B peaks at $N=3$ on MMVet and $N=4$ on ScienceQA), while larger $N$ (≥10 or “All”) degrades accuracy due to long-tail noise and distraction.
>    These findings validate our Top-3 heuristic as a robust and cost-effective choice that balances context coverage and noise suppression. It doesn't cause "context-missing hallucinations". We will include this analysis in the revision.
>    | MMVet| Qwen2-VL-2B|InternVL2-1B|
>    | --- | ---| --- |
>    | 1 obj|72.02|78.44|
>    | 2 obj|68.35|76.61|
>    | 3 obj| **73.85**| 78.44|
>    | 4 obj| 71.10| 78.44|
>    | 5 obj| 69.72| **79.82**|
>    | 10 obj| 68.35| 75.23|
>    | All  obj| 71.10| 77.52|
>    | **ScienceQA** | **Qwen2-VL-2B** | **InternVL2-1B** |
>    | 1 obj| 66.63| 66.98|
>    | 2 obj| 65.89| 67.13|
>    | 3 obj| 65.00| **68.32**|
>    | 4 obj| **67.38**| 68.12|
>    | 5 obj| 66.44| 67.63|
>    | 10 obj| 66.93| 66.88|
>    | All obj| 66.24| 66.88|
>
> > **Q3: Complete Algorithmic Pipeline**
>
> We completely agree with this constructive suggestion. While Section 3 currently describes the process in text, we will add an "Algorithm 1" block in the revision. This will clearly formalize our step-by-step pipeline and greatly enhance both readability and reproducibility.
>
> > **Q4: Statistics on Hard Cases**
>
> This is a valuable suggestion. We define Hard Cases as samples where the reference answer is incorrect but Semantic Entropy (SE) is equal to 0, indicating false consistency. By definition, SE achieves 0% detection accuracy on this subset.
>
> On ScienceQA, DOUBT successfully breaks this false consistency and recovers a substantial portion of these cases across models. For example, it flags 67/165 cases for Qwen2-VL-7B (40.6%) and 88/233 for InternVL2-1B (37.8%).
>
> These results highlight DOUBT’s unique advantage, and we will include this analysis in the revision.
>
> | ScienceQA     | Hard Cases | Successful Cases |
> | ---| --- | --- |
> | Qwen2-VL-2B   | 121| 50|
> | Qwen2-VL-7B   | 165| 67|
> | Qwen2-VL-72B  | 120| 51|
> | InternVL2-1B  | 233| 88|
> | LLaVA-1.5-7B  | 207| 73|
> | LLaVA-NeXT-7B | 159| 64|
>
> > **Q5: Robustness to Diverse Prompt Styles in Stage 2**
>
> This is an insightful suggestion. We evaluate DOUBT with diverse Stage 2 prompts, including concise instructions, CoT (0-shot), and CoT (few-shot).
>
> Results show that DOUBT remains consistently effective across all formats, demonstrating the robustness of the vMF metric. Moreover, prompt diversity can further benefit specific models: Qwen2-VL-2B performs best with CoT (few-shot) (e.g., 74.77% on MMVet), while InternVL2-1B achieves its peak with the original prompt (78.44%).
>
> This indicates that while our default prompt is a strong general baseline, DOUBT can flexibly adapt to different prompt styles for additional gains. We will include this study in the appendix.
>
> | LLaVABench | Qwen2-VL-2B| InternVL2-1B|
> | --- | --- | --- |
> | Instruction| 70.00| 66.67|
> | CoT (0 shot)   | 70.00| 70.00|
> | CoT (few shot) | **73.33**| 61.67|
> | Ours| 68.33| **73.33**|
> | **MMVet** | **Qwen2-VL-2B** | **InternVL2-1B** |
> | Instruction | 71.10| 67.89|
> | CoT (0 shot)   | 68.81| 75.69|
> | CoT (few shot) | **74.77**| 69.72|
> | Ours | 73.85  | **78.44** |

---

> > ### Author Rebuttal · Reviewer_vdrr · 2026-04-02
> >
> > Thanks the author for the comprehensive rebuttal. I'm very satisfied with the additional experiments and evidence to my 4 main concerns addressed above.

---

### Decision · Program_Chairs · 2026-04-30

**Decision:**

Accept (spotlight)

**Comment:**

This paper proposes a black-box hallucination detection framework for MLLMs, which jointly considers object-aware prompting with a vMF-based confidence scoring mechanism.  In the first round, the main concerns relate to methodological clarity, efficiency, and design choices, particularly the use of object-aware prompting (e.g., top-3 object selection), the number of sampling iterations, and the lack of initial discussion on computational cost. The reviewers also commented on more analysis on robustness to prompt variations, embedding model choices, and scalability to complex scenes. Importantly, concerns about the fairness of the comparison and sampling strategy were raised.

After rebuttal, all the concerns were largely resolved through clarifications and additional explanations. All reviewers significantly increased their scores (to Accept / Strong Accept), which indicates that the authors successfully addressed key concerns regarding validity, efficiency trade-offs, and experimental justification. The added clarifications on sampling behavior, hard-case analysis, and prompt robustness notably improved confidence in the proposed method.

Therefore, this paper is viewed as clear, practically relevant, and empirically strong, with a simple yet effective object-aware prompting as well as generalizing across multiple MLLMs. The remaining limitations are mainly about efficiency and design exploration rather than core validity. I recommend accepting this paper, which is the highest score in my batch, and this paper deserves an oral presentation.